# Biological nitrogen fixation of natural and agricultural vegetation simulated with LPJmL 5.7.9

Stephen Björn Wirth[1,2], Johanna Braun[1], Jens Heinke[1], Sebastian Ostberg[1], Susanne Rolinski[1], Sibyll Schaphoff[1], Fabian Stenzel[1], Werner von Bloh[1], Friedhelm Taube[2], and Christoph Müller[1]

[1]Potsdam Institute for Climate Impact Research (PIK), Member of the Leibniz Association, P.O. Box 60 12 03, 14412 Potsdam, Germany
[2]Institute of Crop Science and Plant Breeding, Grass and Forage Science/Organic Agriculture, Kiel University, Hermann-Rodewald-Str. 9, 24118, Kiel, Germany

**Correspondence:** Stephen Björn Wirth (stephen.wirth@pik-potsdam.de)

**Abstract.** Biological nitrogen fixation (BNF) by symbiotic and free living bacteria is an important source of plant-available nitrogen (N) in terrestrial ecosystems supporting carbon (C) sequestration and food production worldwide. Dynamic global vegetation models (DGVMs) are frequently used to assess the N and C cycle under dynamic land use and climate. BNF plays an important role for the components of both these cycles making a robust representation of the processes and variables that BNF depends on important to reduce uncertainty within the C and N cycles and improve the ability of DGVMs to project future ecosystem productivity, vegetation patterns or the land C sink. Still, BNF is often modelled as a function of net primary productivity or evapotranspiration neglecting the actual drivers. We implemented plant functional type-specific limitations for BNF dependent on soil temperature and soil water content as well as a cost of BNF in the Lund Potsdam Jena managed Land (LPJmL) DGVM and compare the new (*C-costly*) against the previous (*Original*) approach and data from the scientific literature. For our comparison we simulated a potential natural vegetation scenario and one including anthropogenic land use for the period from 1901 to 2016 for which we evaluate BNF and legume crop yields. Our results show stronger agreement with BNF observations for the *C-costly* than the *Original* approach for natural vegetation and agricultural areas. The *C-costly* approach reduced the overestimation of BNF especially in hot spots of legume crop production. Despite the reduced BNF in the *C-costly* approach, yields of legume crops were similar to the *Original* approach. While the net C and N balances were similar between the two approaches, the reduced BNF in the *C-costly* approach results in a slight underestimation of N losses from leaching, emissions and harvest compared to literature values, supporting further investigation of underlying reasons, such as processes represented in DGVMs and scenario assumptions. While we see potential for further model development, for example to separate symbiotic and free living BNF, the *C-costly* approach is a major improvement over the simple *Original* approach because of the separate representation of important drivers and limiting factors of BNF and improves the ability of LPJmL to project future C and N cycle dynamics.

# 1   Introduction

Biological nitrogen fixation (BNF) is an important source of plant-available nitrogen (N) in terrestrial ecosystems (Galloway et al., 1995). It can be separated into symbiotic (Granhall, 1981) and free living (Reed et al., 2011) BNF, which account for the total BNF with different shares in different ecosystems (Davies-Barnard and Friedlingstein, 2020b). In natural terrestrial ecosystems, N deposition, N-fixation through lightning, and BNF are the only processes that introduce additional reactive N into the system (Yu and Zhuang, 2020). In agricultural systems, increased N inputs are - together with the extensive manure recycling - a major source of nitrous oxide ($N_2O$) and ammonium ($NH_4^+$) emissions (Reay et al., 2012; Tian et al., 2020) and nitrate ($NO_3^-$) pollution (Moss, 2007). These inputs result from increased BNF and deposition of additional anthropogenic N inputs, which originate mainly from synthetic fertiliser application (Lu and Tian, 2017). Promoting N-fixing crops such as forage and grain legumes for usage as green manure has been discussed (Becker et al., 1995; Fageria, 2007; Northup and Rao, 2016) to reduce N losses from nitrification, volatilization, denitrification and leaching on agricultural land. Generally, symbiotic as well as free living BNF can be important for plant growth in N limited ecosystems and supports carbon (C) sequestration and food production across the globe.

Briefly summarised, BNF describes the transformation of atmospheric $N_2$ to ammonia ($NH_4^+$) by a variety of soil microorganisms providing a source of mineral N for plants at the expense of C (Yu and Zhuang, 2020). The underlying mechanisms of BNF as well as its role within the C and N cycles and for ecosystem productivity have been described in detail in multiple studies (e.g., Yu and Zhuang, 2020; Davies-Barnard and Friedlingstein, 2020a; Cleveland et al., 1999). Here, we focus on the representation of BNF in the Lund Potsdam Jena managed Land (LPJmL) DGVM (Schaphoff et al., 2018b; von Bloh et al., 2018; Lutz et al., 2019; Herzfeld et al., 2021; Porwollik et al., 2022; Heinke et al., 2023). We do not distinguish between symbiotic and free living BNF throughout this study but only consider total BNF as the sum of both forms.

DGVMs such as LPJmL can be used to assess the role of BNF for the productivity of natural and agricultural ecosystems and its effects on the N and C cycle under dynamic land use and climate. A solid representation of the processes behind BNF is important to reduce uncertainty and improve model results of DGVMs, which are frequently used in impact assessments and to inform policy makers. A variety of approaches of different complexity to model BNF have been developed. A key difference between approaches is the selection of variables that control BNF and the accounting of the C cost of BNF. For example, Cleveland et al. (1999) use actual evapotranspiration as a single explanatory variable, while Yu and Zhuang (2020) consider soil temperature, soil water content, soil mineral N and soil C content. Both these approaches do not consider the cost of BNF neglecting the reduced C assimilation (Cleveland et al., 1999; Yu and Zhuang, 2020), while others explicitly consider a cost per amount of N fixed and a maximum amount of C that can be invested in BNF (e.g., Ma et al., 2022). Even more complex approaches consider the different pathways of N uptake that are associated with a cost (active N uptake, retranslocation and BNF) and optimise for the minimum cost (e.g., Fisher et al., 2010). Depending on the considered variables, the simulated BNF and how it is affected by climate change may strongly differ, which in turn can have strong effects on the simulated C and N fluxes and pools.

A comparison to data published by Davies-Barnard and Friedlingstein (2020a) suggests that the approach that was implemented in LPJmL (von Bloh et al., 2018) based on Cleveland et al. (1999) - in the following defined as the *Original* approach - overestimates global BNF. In addition, we identified several shortcomings of the *Original* approach in LPJmL: In the *Original* approach, BNF is a function of actual evapotranspiration, which leads to an overestimation of BNF in moist but not necessarily N-limited ecosystems and an underestimation in dry but N-limited ecosystems. In this simplified implementation, BNF is not constrained by the availability of reactive forms of N and additional N is fixed even if the reactive soil N is sufficient to fulfil the N demand, which potentially leads to an overestimation of the ammonia pool and N losses. For cultivated grain legumes, the approach assumes no limitation of BNF at all but simply supplies all N requested by the plant that cannot be fulfilled through N uptake from mineral N pools in the soil. This leads to an overestimation of cropland BNF. In order to overcome these deficiencies, we here describe a revision of the *Original* approach in LPJmL with a more complex approach, referred to as *C-costly* approach in the following. The *C-costly* approach is inspired by Ma et al. (2022) and Yu and Zhuang (2020) and introduces plant functional type (PFT)-specific limitations for BNF dependent on soil temperature and soil water content as well as a C cost of BNF. In the following, we present the *C-costly* BNF approach and evaluate its performance against global and site-specific data. We discuss the differences between the *Original* and the *C-costly* BNF approach for the N-cycle and plant productivity.

## 2 Methods

### 2.1 Model description

LPJmL is a dynamic global vegetation model (DGVM) with the full terrestrial hydrology and explicit representation of agricultural management systems for cropland and pastures. We have implemented the BNF module in the most recent development branch, which is based on a consolidated version of the carbon-only model (LPJmL4, Schaphoff et al., 2018b, a), the N cycle (LPJmL5, von Bloh et al., 2018), tillage (Lutz et al., 2019), manure (Herzfeld et al., 2021), cover crop (Porwollik et al., 2022), and grazing management (Heinke et al., 2023) modules. There have been further model improvements that have not been described in publications elsewhere, including improved online coupling options with other models such as IMAGE (Müller et al., 2016) or copan:CORE (Donges et al., 2020). For a better representation of crops that are not explicitly represented (referred to as *others*), these are no longer assumed to be identical to managed grassland (Bondeau et al., 2007), but can be simulated as separate stands with distinct management inputs (e.g. fertiliser amounts).

The original spinup protocol for LPJmL4, described in Schaphoff et al. (2013), was modified to account for the interaction between soils and plants through N supply in LPJmL5. The principal technique to accelerate the spinup by calculating the equilibrium soil C stocks from litter decomposition (i.e., the flux of C into the soil C pools) and soil C turnover rates (or residence time) remains the same as in Schaphoff et al. (2013). However, the original code was refactored to improve the accuracy of estimates of equilibrium stocks and to apply the technique to soil C and N pools simultaneously.

In LPJmL5, an adjustment of N pools can lead to a change in plant productivity through a change in N supply from mineralisation. To account for this feedback, the C- and N-stock adjustments need to be repeated multiple times until the soil and

the vegetation reach equilibria. The revised spinup procedure starts with an initial period of 300 years during which vegetation is allowed to establish. This is followed by a 2400-year period, during which soil C and N pools are updated every 15 years based on litter decomposition and soil pool turnover rates of the preceding ten years. This long period with repeated adjustment (160 times) of C and N pools is required to reach an equilibrium in regions with very low turnover rates (e.g., in the boreal zone). To reduce the effect of inter-annual variability on estimates of equilibrium stocks, a final adjustment is applied after 300 simulation years using litter decomposition and soil pool turnover rates over that period. Finally, the model is allowed to adjust to the new C and N stocks for another 500 simulation years.

To assess the effectiveness of the spinup procedure, we conducted a 1000-year model run under the same conditions as during the spinup period (i.e., stable pre-industrial atmospheric $CO_2$ concentration, and atmospheric N deposition, and climate) for which we present results in appendix C.

Further changes to the code since the last published version (see Porwollik et al., 2022) include various bug fixes concerning fertiliser and manure application, data output, environmental flow requirements (Jägermeyr et al., 2017), soil temperature (Schaphoff et al., 2013), and bioenergy plantations (Beringer et al., 2011). Latest code changes are now also documented in a `CHANGELOG.md` file as part of the code repository (section 5).

## 2.2 BNF-relevant nitrogen cycle components in LPJmL

While we refer to von Bloh et al. (2018) for a detailed description and evaluation of the N cycle in LPJmL, we briefly describe the main processes that determine N deficit - which is the prerequisite for N-fixation in the *C-costly* approach - and the *Original* approach here and provide the full equations in appendix A. N deficit is defined as the difference between the plant N demand (Eq. A4) and the active and passive N uptake (Eq. A5) and labile N reserves (Eq. A11).

$$N_{\mathrm{deficit,t}} = N_{\mathrm{demand,t}} - (N_{\mathrm{uptake,t}} + N_{\mathrm{labile,t}}) \tag{1}$$

The N demand accounts for N required to produce RuBisCo depending on the maximum carboxylation capacity and the leaf area index (LAI) of the respective PFT (Eq. A1 first summand) and the structural N demand depending on the current N content of the different plant compartments (Eq. A1 second summand and A4). N reserves are included using a PFT-specific parameter (Eq. A4).

The N uptake is calculated as a combination of passive and active N uptake from the soil and is a function of the potential N uptake of the root system (Eq. A5) which is reduced to account for soil mineral N availability (Eq. A8), soil temperature (Eq. A9) and plant N starvation (Eq. A10). Labile N reserves represent the N currently available from past N uptake, BNF or retranslocation (Eq. A11).

In the *Original* approach, BNF was calculated from the 20 year average of annual evapotranspiration ($etp$) for tree and herbaceous PFTs following the function from Cleveland et al. (1999):

$$\mathrm{BNF} = \begin{cases} \max(0, (0.0234 \cdot \mathrm{etp} - 0.172)/10/365) & \text{if } C_{\mathrm{root}} > 20 \ gC \ m^{-2} \\ 0 & \text{otherwise} \end{cases} \tag{2}$$

The resulting BNF is added to the $NH_4^+$ pool of the first soil layer. For crop PFTs, BNF equals $N_\text{deficit}$ and is directly added to $N_\text{labile}$.

## 2.3 The *C-costly* approach

A key feature is the connection of BNF to an associated cost represented as a reduction of net primary production (NPP). The *C-costly* approach calculates actual BNF ($N_{fix}$) from the potential BNF ($N_\text{fix,pot}$) using several reduction factors. First, the N fixation rate for the environmental conditions $N_\text{fix,env}$ is calculated from $N_\text{fix,pot}$ for the first two soil layers $l = 1, 2$ accounting for reductions by dimensionless soil temperature and soil water content (SWC) limitations functions ($f_T$, $f_W$) and the root distribution $\mathrm{rootdist}$ in the interval $[0, 1]$(Ma et al., 2022):

$$N_\text{fix,env} = \sum_{l=1}^{2} N_\text{fix,pot} \cdot f_T(T_\text{soil,l}) \cdot f_W(\mathrm{SWC}_l) \cdot \mathrm{rootdist}_l \tag{3}$$

The soil temperature limitation is increasing linearly outside the optimal temperature interval $[T_\text{opt,low}, T_\text{opt,high}]$, Eq. 4, Fig. 1 a) and prohibits BNF if outside the tolerable temperature interval $[T_\text{min}, T_\text{max}]$, while the soil water limitation is linearly dependent on the relative soil water content SWC (Eq. 5, Fig. 1 b).

$$f_T(T_\text{soil}) = \begin{cases} 0, & \text{if } T_\text{soil} < T_\text{min} \text{ or } T_\text{soil} > T_\text{max} \\ \frac{T_\text{soil} - T_\text{min}}{T_\text{opt,low} - T_\text{min}}, & \text{if } T_\text{min} \leq T_\text{soil} < T_\text{opt,low} \\ 1, & \text{if } T_\text{opt,low} \leq T_\text{soil} \leq T_\text{opt,high} \\ \frac{T_\text{max} - T_\text{soil}}{T_\text{max} - T_\text{opt,high}}, & \text{if } T_\text{opt,high} < T_\text{soil} \leq T_\text{max} \end{cases} \tag{4}$$

$$f_W(\mathrm{SWC}) = \begin{cases} 0, & \text{if } \mathrm{SWC} \leq \mathrm{SWC}_\text{low} \\ \varphi_1 + \mathrm{SWC} \cdot \varphi_2, & \text{if } \mathrm{SWC}_\text{low} < \mathrm{SWC} < \mathrm{SWC}_\text{high} \\ 1, & \text{if } \mathrm{SWC} \geq \mathrm{SWC}_\text{high} \end{cases} \tag{5}$$

The root distribution is calculated as in Eq. A7. Since only the fraction of roots in the first two soil layers is used for BNF, shallow root profiles lead to a higher BNF compared to deep root profiles. $N_\text{fix,pot}$, $T_\text{min}$, $T_\text{opt,low}$, $T_\text{opt,high}$, $T_\text{max}$, $\mathrm{SWC}_\text{low}$, $\mathrm{SWC}_\text{high}$, $\varphi_1$ and $\varphi_2$ are PFT-specific parameters (Tab. 1) and their values are adopted from Yu and Zhuang (2020) for the natural vegetation PFTs and from Ma et al. (2022) for soybean and pulses.

If $N_\text{fix,env}$ exceeds the amount of N missing to fulfil the N demand of the current day (the N deficit $N_\text{deficit}$), the N fixation is reduced:

$$N_\text{fix,need} = \min(N_\text{deficit}, N_\text{fix,env}) \tag{6}$$

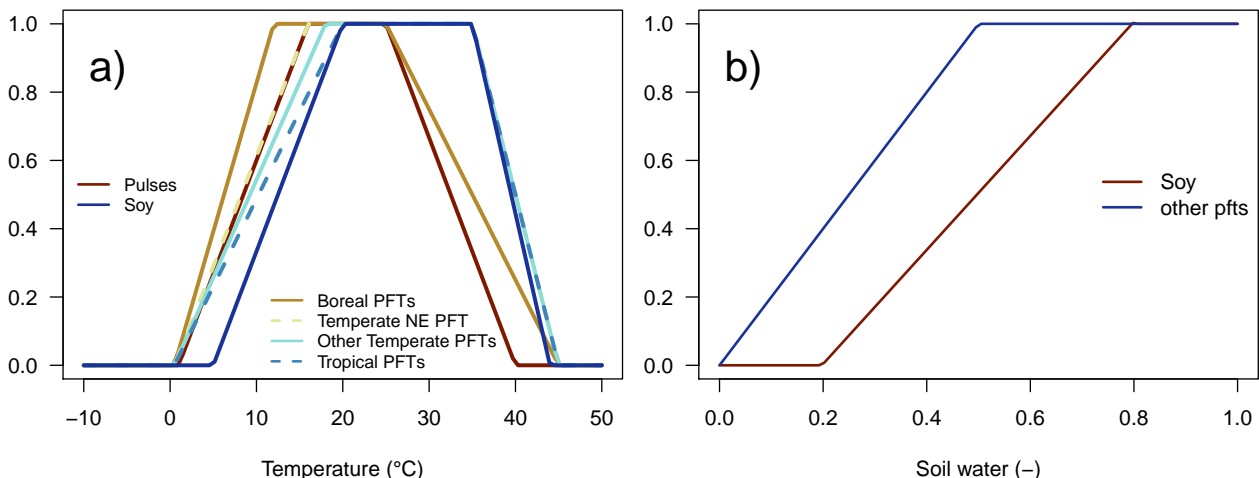

**Figure 1.** Dimensionless temperature limitation function $f_T(T)$ (a) and soil water limitation function $f_W(\text{SWC})$ (b)

Finally, if the cost of N fixation exceeds the NPP available for BNF, N fixation is further reduced to match the maximum amount that can be fixed with the current day's NPP share available for BNF.

$$N_{\text{fix}}(\text{NPP}) = \begin{cases} N_{\text{fix,need}}, & \text{if } cost_{\textbf{BNF}} \cdot N_{\text{fix,need}} < f_{\text{fixer}} \cdot f_{\textbf{NPP}} \cdot \text{NPP} \\ f_{\text{fixer}} \cdot f_{\textbf{NPP}} \cdot \text{NPP}/cost_{\textbf{BNF}}, & \text{otherwise,} \end{cases} \tag{7}$$

where $f_{\textbf{NPP}}$ is the maximum share (dimensionless) of NPP available for BNF, which is set to 0.14 (Kull, 2002) for the natural PFTs and to 0.25 for soybean and pulses. The average N fixer fraction ($f_{\text{fixer}}$) is set to 0.05 for the tropical, to 0.01 for the

temperate, and to 0.03 for the boreal zone (Yu and Zhuang, 2020). PFTs only fix additional N if the N uptake from other sources is insufficient and the net primary productivity (NPP) is larger than zero. The costs of BNF are set at a moderate constant value of 6 gC gN$^{-1}$ (Boote et al., 2009; Ryle et al., 1979; Patterson and Larue, 1983; Kaschuk et al., 2009).

### 2.4   Modelling protocol

To compare the two BNF approaches, we simulated two scenarios: First, a potential natural vegetation (PNV) scenario, which

does not include anthropogenic land use or agricultural production systems. Second, a scenario that includes agricultural land use (LU). The same input datasets were used for all scenarios. We used the climate data from the GSWP3-W5E5 dataset (Kim; Cucchi et al., 2020; Lange et al., 2022), historical atmospheric N deposition (Yang and Tian, 2020), historical atmospheric CO$_2$ concentrations (Büchner and Reyer, 2022), historical land-use patterns (Ostberg et al., 2023) and grazing management data (Stenzel et al., 2023). For both BNF approaches, we conducted spinup simulations of 3500 years using a random permutation

of the climate data from 1901 to 1930. These spinup simulations ensure that the C and N balances are in an equilibrium. Afterwards, land use is introduced and a second spinup period of 390 years is run to capture the effects of historical land-use

**Table 1.** BNF related PFT-specific parameter values for the tropical broadleaved evergreen tree (TrBE), tropical broadleaved raingreen tree (TrBR), temperate needleleaved evergreen tree (TeNE), temperate broadleaved evergreen tree (TeBE), temperate broadleaved summergreen tree (TeBS), boreal needleleaved evergreen tree (BoNE), boreal broadleaved summergreen tree (BoBS), boreal needleleaved summergreen tree (BoNS), tropical herbaceous (TrH), temperate herbaceous (TeH), polar herbaceous (PoH), soybean and pulses.

| PFT | $N_{\text{fix,pot}}$ $\text{gNm}^{-2}\text{d}^{-1}$ | $T_{\min}$ $^\circ$C | $T_{\text{opt,low}}$ $^\circ$C | $T_{\text{opt,high}}$ $^\circ$C | $T_{\max}$ $^\circ$C | $\text{SWC}_{\text{low}}$ $\text{m}^3\text{m}^{-3}$ | $\text{SWC}_{\text{high}}$ $\text{m}^3\text{m}^{-3}$ | $\varphi_1$ - | $\varphi_2$ - | $f_{\text{NPP}}$ - | $\text{cost}_{\text{BNF}}$ $\text{gCg}^{-1}\text{N}$ | $f_{\text{fixer}}$ - |
|---|---|---|---|---|---|---|---|---|---|---|---|---|
| TrBE | 0.01 | 0.5 | 20 | 35 | 45 | 0 | 0.5 | 0 | 2.0 | 0.14 | 6 | 0.05 |
| TrBR | 0.01 | 0.5 | 20 | 35 | 45 | 0 | 0.5 | 0 | 2.0 | 0.14 | 6 | 0.05 |
| TeNE | 0.01 | 0.5 | 16 | 35 | 45 | 0 | 0.5 | 0 | 2.0 | 0.14 | 6 | 0.01 |
| TeBE | 0.01 | 0.5 | 18 | 35 | 45 | 0 | 0.5 | 0 | 2.0 | 0.14 | 6 | 0.01 |
| TeBS | 0.01 | 0.5 | 18 | 35 | 45 | 0 | 0.5 | 0 | 2.0 | 0.14 | 6 | 0.01 |
| BoNE | 0.01 | 0.5 | 12 | 25 | 45 | 0 | 0.5 | 0 | 2.0 | 0.14 | 6 | 0.03 |
| BoBS | 0.01 | 0.5 | 12 | 25 | 45 | 0 | 0.5 | 0 | 2.0 | 0.14 | 6 | 0.03 |
| BoNS | 0.01 | 0.5 | 12 | 25 | 45 | 0 | 0.5 | 0 | 2.0 | 0.14 | 6 | 0.03 |
| TrH | 0.01 | 0.5 | 20 | 35 | 45 | 0 | 0.5 | 0 | 2.0 | 0.14 | 6 | 0.05 |
| TeH | 0.01 | 0.5 | 18 | 35 | 45 | 0 | 0.5 | 0 | 2.0 | 0.14 | 6 | 0.01 |
| PoH | 0.01 | 0.5 | 12 | 25 | 45 | 0 | 0.5 | 0 | 2.0 | 0.14 | 6 | 0.03 |
| Soybean | 0.1 | 5 | 20 | 35 | 44 | 0.2 | 0.8 | -0.33 | 1.67 | 0.25 | 6 | 1 |
| Pulses | 0.1 | 1 | 16 | 25 | 40 | 0 | 0.5 | 0 | 2.0 | 0.25 | 6 | 1 |

change on the C and N cycle. Following the two spinup simulations, the model is run from 1901 until 2016 using the transient input data.

## 2.5 Model evaluation

We compared simulated total global BNF for both approaches against several estimates, which were derived empirically or reported in other modelling studies. Data on these estimates are available from Davies-Barnard and Friedlingstein (2020a). The global BNF is calculated as the sum of BNF per area times grid cell area over all grid cells:

$$\text{BNF}_{\text{glob}} = \sum_{\text{cell}}^{n_{\text{cell}}} \text{BNF}_{\text{cell}} \cdot \text{area}_{\text{cell}} \tag{8}$$

For the evaluation we calculate the median, minimum and maximum between 2001 and 2010 and qualitatively compare these
values against past estimates. We calculated the overlap between our results and reported data if minimum and maximum

values were available.

$$\text{overlap} = \begin{cases} 0 & \text{if } x_{\min} > y_{\max} \text{ or } x_{\max} < y_{\min} \\ (\min(x_{\max}, y_{\max}) - \max(x_{\min}, y_{\min}))/(y_{\max} - y_{\min}) & \text{otherwise} \end{cases}, \tag{9}$$

where $x_{\min}$ and $x_{\max}$ are the simulated minimum and maximum and $y_{\min}$ and $y_{\max}$ are the minimum and maximum values from the literature.

In addition, we compared our results to data obtained at several sites for the natural vegetation (Davies-Barnard and Friedlingstein, 2020b) and legume crops (Ma et al., 2022). To evaluate legume crop BNF and yields, we conducted additional local simulations matching the coordinates of the experiments following the protocol described in sect. 2.4 but ensured that the respective crops (soybean or pulses) were grown under the reported water management (rainfed or irrigated). We calculated the root mean square error (RMSE) as follows:

$$\text{RMSE} = \sqrt{\sum_{n}^{N} (x_n - y_n)^2/N}, \tag{10}$$

where $N$ is the number of observations and $x_n$ and $y_n$ are the simulated and observed values.

## 3 Results

### 3.1 Comparison of the BNF approaches

Comparing the simulated BNF of both approaches to data from literature and experiments showed substantial improvement of the global BNF (sect. 3.1.1) as well as the latitudinal and spatial patterns (sect. 3.1.2).

#### 3.1.1 Comparison to data and other models

The two approaches show large differences in the simulated BNF. While the median global BNF between 2001 and 2010 was 191 TgNyr$^{-1}$ for the *Original* approach, for the *C-costly* approach it was substantially lower with a value of 109 TgNyr$^{-1}$ (Fig. 2 a). Comparing the global BNF of both approaches to estimates from the scientific literature shows agreement of the *C-costly* values with several other data sources, while the *Original* approach overestimates most of the literature values. Especially the recent estimate by Davies-Barnard and Friedlingstein (2020a) was closely matched by the *C-costly* approach and 60% of the simulated data were within the range of the Davies-Barnard and Friedlingstein (2020a) data (Fig. 2 a). Despite the fact that the *Original* approach was not derived from the empirical relationship of Cleveland et al. (1999) for the legume crops, the data from Cleveland et al. (1999) are well matched by the *Original* approach and only the spread of the Cleveland et al. (1999) data is underestimated. In comparison to the data of Xu-Ri and Prentice (2017), who reported much higher values compared to the other studies, BNF is underestimated by both approaches implemented in LPJmL. However, large differences are to be expected considering that their approach does not calculate the actual BNF but rather the BNF needed to sustain global NPP (Xu-Ri and Prentice, 2017).

Comparing the spatial patterns of the two approaches to those of Davies-Barnard and Friedlingstein (2020a) shows that the *Original* approach generally overestimated BNF in large areas of the tropics and temperate zones (Fig. C4 c). The *C-costly* approach still overestimates BNF in the tropics and the production areas of soybean and/or pulses in India and the United States of America (USA) but values are substantially smaller than in the *Original* approach (Fig. C4 f). In both approaches, observed BNF is slightly underestimated in the central to western part of the USA, Canada, China, Kazakhstan, Russia and Mongolia.

On croplands, BNF was 21 $\mathrm{TgN}\ \mathrm{yr}^{-1}$ with the *C-costly* approach, which is within the range of 17 to 31 $\mathrm{TgN}\ \mathrm{yr}^{-1}$ reported by a recent review (Zhang et al., 2021) and other studies (Bodirsky et al., 2012; Chang et al., 2021). This contrasts the overestimation of cropland BNF in the *Original* approach which was 68 $\mathrm{TgN}\ \mathrm{yr}^{-1}$. For the two legume crop functional types soybean and pulses, we compared the simulation results to BNF and yield data from experiments (Fig. 2b and c and Fig. C1 a and b). For all except two experiments, the *Original* approach strongly overestimated BNF independent of the crop and the irrigation management. Using the *C-costly* approach, the cropland BNF was strongly reduced by a factor of approximately two leading to substantially lower RMSEs. While simulation results were closer to observations, some deviations remain. Pulses generally showed lower BNF for both approaches compared to soybean, while irrigated simulations generally showed a higher BNF and overestimated BNF compared to observations for all experiments in the *Original* and for the vast majority in the *C-costly* approach. Crop yields barely differed between the two approaches and were comparable to observations (Fig. C1 a and b).

### 3.1.2 Global variation in BNF

Generally, BNF decreases from low to high latitudes with similar gradients but from different levels for the two approaches (Fig. 3). In latitudes with a high share of crop legumes (e.g. 30 to 40°S) the reduction of BNF in the *C-costly* approach is especially large. Both the *Original* as well as the *C-costly* approach underestimate BNF at high latitudes (the *Original* more strongly so) compared to Davies-Barnard and Friedlingstein (2020a). The *C-costly* approach shows good performance in the mid latitudes, but both approaches overestimate BNF compared to observations in the tropics (Fig. 3). In the *Original* approach, especially the high BNF of cropland contributes to the overestimation. For the low latitudes, both approaches exceed the values from Davies-Barnard and Friedlingstein (2020b). However, the higher BNF in the tropics is comparable to the median of the TRENDY-N ensemble (sect. 4 and Kou-Giesbrecht et al., 2023).

With the *Original* approach, mineral N was added to the first soil layer and subsequently incorporated by the PFTs via the passive and active N uptake pathway. This did not allow a separate identification of N taken up via BNF from the total N uptake except for the legume crops which fixed their entire N deficit. Using the *C-costly* BNF, the model separates N uptake by BNF from passive and active N uptake against N concentration gradients (Marschner et al., 1991; Fisher et al., 2010)) facilitating the analysis of the share of BNF in total N uptake subsequently referred to as $\%N_{\mathrm{dfa}}$ which is commonly used to refer to this variable in the empirical literature (e.g. Herridge et al., 2008). In the PNV simulation, values for $\%N_{\mathrm{dfa}}$ were between 0 and 20% for most of the grid cells (Fig. C7 and C8 b). The distribution is bimodal showing a peak below 5% and one at approximately 10%. For the dynamic land-use simulation, the values for $\%N_{\mathrm{dfa}}$ are similar but the second peak is barely distinguishable because of a higher share of $\%N_{\mathrm{dfa}}$ values between 5 and 10% (Fig. C7 and C8 a). For the crop legumes, $\%N_{\mathrm{dfa}}$ is substantially higher, with the peak around 80% (Fig. C7 d). The highest values are simulated at low latitudes, especially in

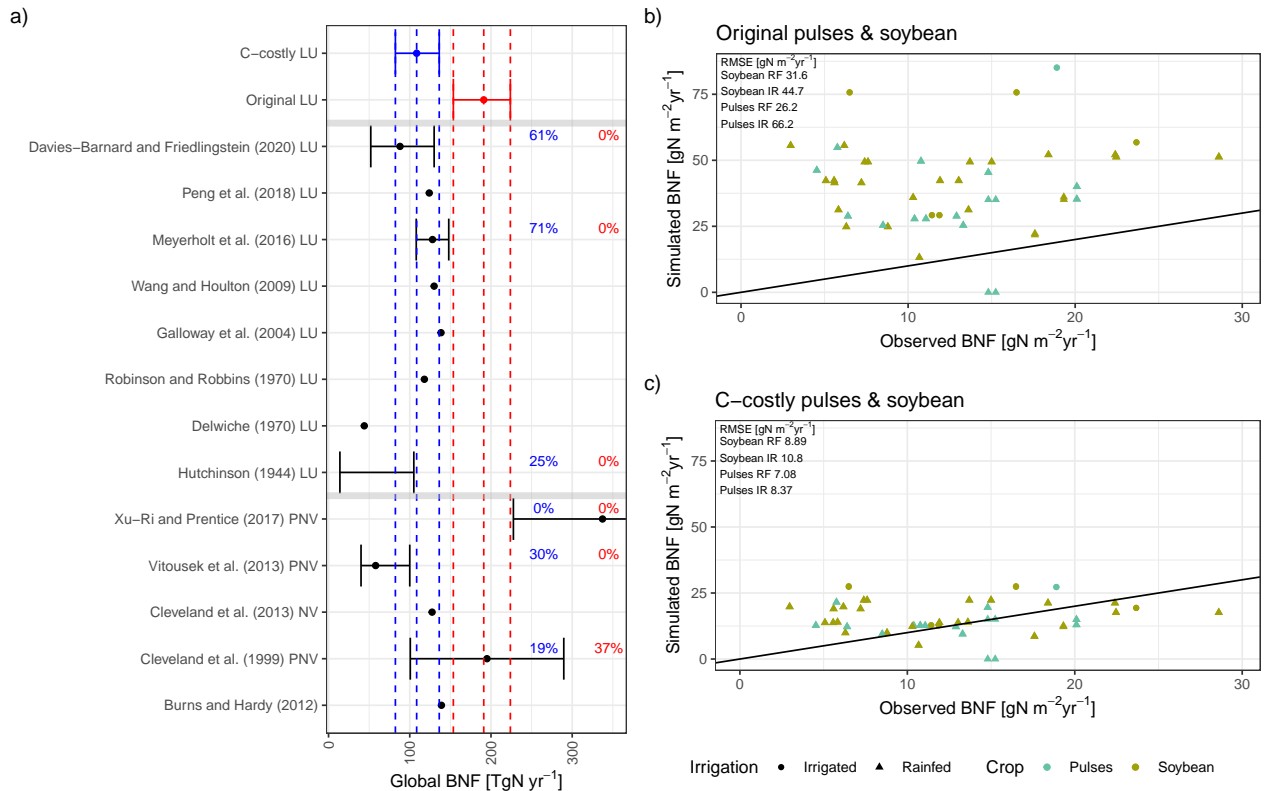

**Figure 2.** Evaluation against global (a) and site specific data (b, c). Global evaluation plot inspired by Davies-Barnard and Friedlingstein (2020a) showing global BNF in $\text{TgN yr}^{-1}$ from different studies (black) compared to the *Original* (red) and *C-costly* (blue) BNF approach implemented in LPJmL. Studies are labelled by author names and whether they consider potential natural vegetation (PNV), actual natural vegetation (NV) or actual land use (LU). We assigned the Davies-Barnard and Friedlingstein (2020a) data to the LU category because they consider cropland area as grasslands and not as potential forest areas. Percentage values give the overlap between the ranges of the simulation results and the literature estimates derived using Eq. 9. Simulated values are the median between 2001 and 2010 and ranges show minimum and maximum. Site specific evaluation (b, c) comparing data from observations for soybean (green) and pulses (blue) for rainfed (RF) (circle) and irrigated (IR) (triangle) experiments and simulations results using the *Original* (b) and *C-costly* (c) BNF approach. Labels show the RMSE of the two approaches.

India, Sub-Sahara Africa, and South America, while the lowest values are simulated in Canada, Russia and southern China (Fig. C8 d). In the *Original* approach $\%N_{\mathrm{dfa}}$ of legume crops was almost exclusively 100% (Fig. C7 e).

In both approaches, BNF per area is higher for agricultural land than for natural vegetation (Fig. C3 d and f). BNF is especially high in hot spots of legume crop production such as Argentina, Brazil, India and the USA (Fig. 3 a and b). While the spatial pattern is similar between the two approaches, in the *Original* approach, the cropland BNF leads to prominent peaks in the latitudinal distribution (Fig. 3 c). These peaks correspond to hot spots of legume crop production where the *C-costly* approach is up to 15 gN m$^{-2}$ yr$^{-1}$ lower (Fig. C4).

For natural vegetation, the differences are smaller. Here, the BNF in the *Original* approach is up to 4 gN m$^{-2}$ yr$^{-1}$ higher compared to the *C-costly* approach (Fig. C4). Here, the spatial patterns differ and show a stronger reduction of BNF in dry regions (e.g. central Australia, the Eurasian steppe regions, south east China and parts of Africa).

The various natural PFTs contribute differently to the lower overall BNF in the *C-costly* approach (Fig. C5 and C6). To some extent this reflects changes in the PFT distribution (Fig. S1 and 2). For the tropical PFTs, BNF is lower for the broadleaved rain-green tree ($\Delta 5.25$ TgN yr$^{-1}$ Fig. C5 b) and the herbaceous PFT ($\Delta 14.1$ TgN yr$^{-1}$ Fig. C5 i) and higher for the broadleaved evergreen tree ($\Delta 7.3$ TgN yr$^{-1}$ Fig. C5 a). While the temperate needleleaved evergreen tree PFT contributed to BNF in low latitudes outside its expected habitat (e.g. in India and Brasil) in the *Original* approach, this issue was resolved with the *C-costly* approach (Fig. C5 c). The temperate PFTs all fix less N in the *C-costly* approach than in the *Original* approach. The reductions are smaller for the broadleaved evergreen ($\Delta 3.6$ TgNyr$^{-1}$ Fig. C5 d) and summergreen ($\Delta 3.8$ TgNyr$^{-1}$ Fig. C5 e) tree and the herbaceous PFT ($\Delta 4.7$ TgNyr$^{-1}$ Fig. C5 j) compared to the needleleaved evergreen tree ($\Delta 9.1$ TgNyr$^{-1}$ Fig. C5 c). The boreal PFTs' BNF is similar ($\Delta$ around $0.5$TgNyr$^{-1}$ Fig. C5 f,g,k) for all PFTs except the needleleaved summergreen tree ($\Delta 1.2$ TgNyr$^{-1}$ Fig. C5 h), which fixes less N in the *C-costly* approach. In the *Original* approach, the temperate herbaceous contributed twice as much as in the *C-costly* to the biological N fixation of the polar vegetation (Fig. C5 j). For the pulses, the BNF was $14.6$ TgNyr$^{-1}$ and for soybean $6.4$ TgNyr$^{-1}$ lower with the *C-costly* approach.

## 3.2 Effects on the nitrogen and carbon cycle and productivity

In LPJmL the C and N cycles are coupled via, for example, the N limitation of gross primary productivity (GPP), which controls the amount of assimilated C, the role of plant organ carbon-to-nitrogen (C:N) ratios for maintenance respiration and the availability of the resulting NPP for BNF. Additionally, the N content of the different plant organs (leaves, roots, sapwood, heartwood and storage organs) is derived dependent on the respective C content ensuring that their C:N ratios remain within a prescribed range. As a result, the N balance components presented in the following section are strongly shaped by their C cycle counterparts as the overall C and N balances represented by LPJmL are intimately linked.

We describe the N balance as the sum over in- and outfluxes of the vegetation and the soil. Therefore, the overall balance contains a change in vegetation and soil N stocks including organic and mineral forms of N.

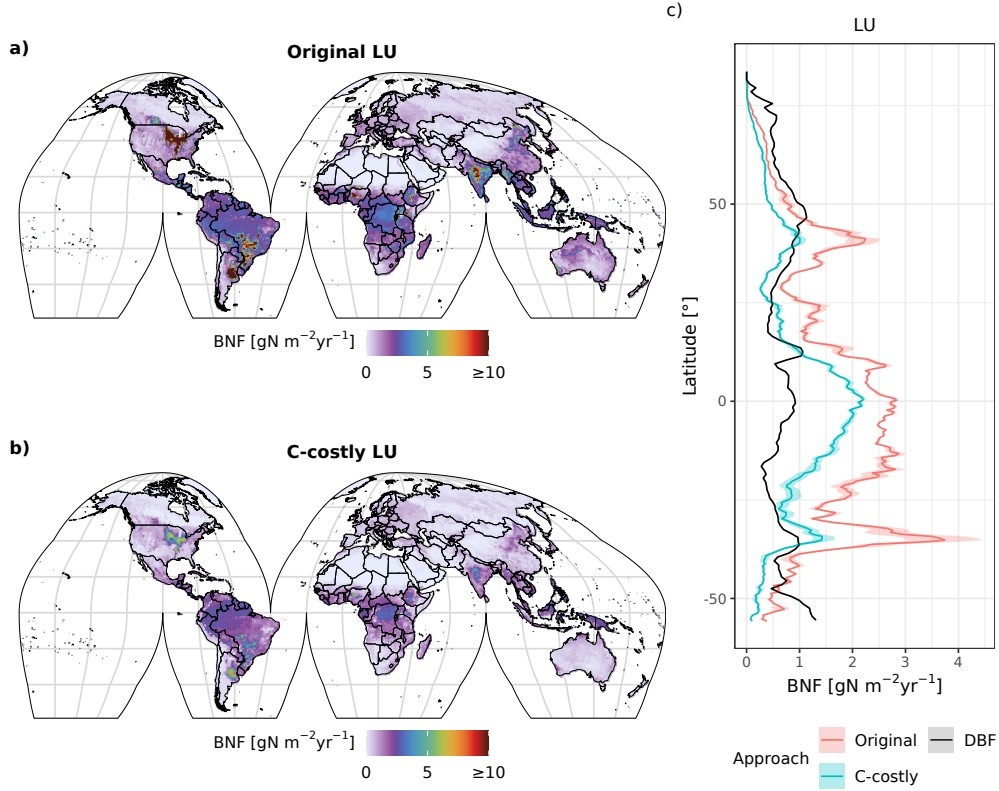

**Figure 3.** Simulated average annual BNF in $gNm^{-2}yr^{-1}$ for years 2001 to 2010 using the *Original* (a) and *C-costly* (b) approach. Average (line) and 5th to 95th percentile (shading) of simulated and observed BNF per latitude in $gNm^{-2}yr^{-1}$ using the *Original* (red), *C-costly* (blue) approach and data from Davis-Barnard & Friedlingstein 2021 (DBF) (c).

### 3.2.1   Potential natural vegetation

Simulating only natural vegetation resulted in a positive terrestrial N balance with an average sink of 52 $TgN\ yr^{-1}$ for the *Original* and 54 $TgN\ yr^{-1}$ for the *C-costly* approach between 2001 and 2010 (Fig. 4 a, b and Tab. 2). In 1901, N in- and outputs were almost balanced and the sink remained small until the 1950s when N inputs from deposition increased resulting in an increased sink. While the overall N balance was similar for both BNF approaches, the size of several components was different. The total BNF simulated with the *Original* approach was approximately double that of the *C-costly* BNF leading to

higher soil mineral N and organic C and N stocks. However, mineral N stocks were not utilised by the vegetation but instead lost to the atmosphere and water bodies leading to higher N emissions and leaching using the *Original* approach. Here, 112 $TgN\ yr^{-1}$ were emitted and 56 $TgN\ yr^{-1}$ were leached on average between 2001 and 2010, while for the *C-costly* approach only 79 $TgN\ yr^{-1}$ were emitted and 39 $TgN\ yr^{-1}$ were leached (Tab. 2). All types of emissions are lower with the *C-costly* approach. $NH_3$ emissions from volatilization decrease by 14 $TgN\ yr^{-1}$, $N_2$ emissions by 12 $TgN\ yr^{-1}$, $N_2O$ emissions by 3

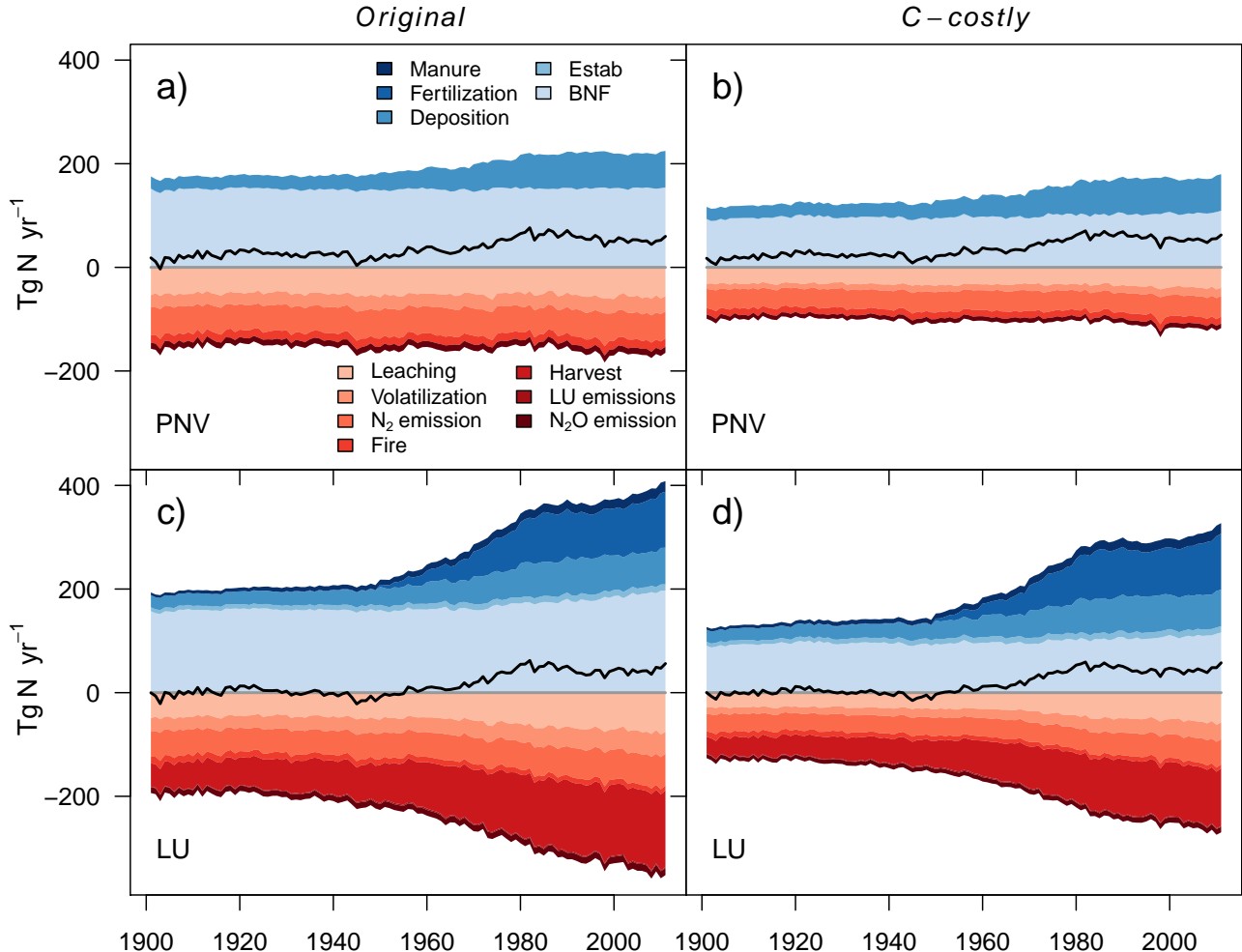

**Figure 4.** Global terrestrial N balance. Scenarios include the *Original* approach, the *C-costly* approach for natural vegetation and actual land use. Net balance is denoted by the black line. N inputs include N from manure, synthetic fertilisers deposition, PFT establishment (Estab) and BNF. N losses include leaching, volatilization, $N_2$ emissions, fire N, harvested N, land-use change emissions (= deforestation and product turnover) and $N_2O$ emissions from nitrification and denitrification.

$TgN\ yr^{-1}$ and fire emissions by 5 $TgN\ yr^{-1}$. Synchronised with the increase of deposition over time, emissions and leaching also increase in both approaches with stronger increases in the *C-costly* approach. Overall, N inputs increased by 35 TgN in total from 1950 to 2000 in the *Original* approach and by 42 TgN in the *C-costly* approach, while N losses from emissions and leaching increased by 1 TgN and 4 TgN respectively (Tab. 2).

     In addition to the changes of several N cycle components, we excepted changes of C cycle components. Overall, the C

balance was similar for both approaches (Fig. C9 a and b). For the PNV simulations, the only C input into the system was the

**Table 2.** N balance values for 2001 to 2010 shown in figures. LUC includes deforestation emissions, product turnover and negative N fluxes

| | *Original* | *C-costly* | Literature | *Original* PNV | *C-costly* PNV | Literature |
|---|---|---|---|---|---|---|
| **N** losses (TgN yr$^{-1}$) | 344 | 263 | | 168 | 118 | |
| Leaching | 74 | 55 | 93[1], 68[2] | 56 | 39 | 28.6[3] |
| Volatilization | 43 | 32 | 21.4[4,5] | 31 | 17 | - |
| N$_2$ emissions | 60 | 47 | 68[1],64.2[2] | 52 | 40 | - |
| N$_2$O emissions | 13 | 10 | 10.9[6],13[7],10[8],7.4-12.3[9],12.9[10] | 12 | 9 | - |
| Fire | 10 | 8 | - | 17 | 13 | - |
| Harvest | 142 | 108 | - | 0 | 0 | - |
| LUC | 2 | 2 | - | 0 | 0 | - |
| **N** gains (TgN yr$^{-1}$) | 388 | 307 | - | 220 | 172 | - |
| BNF | 191 | 110 | see Fig. 2 | 153 | 104 | 19.8-107.9[11] |
| Establishment fluxes | 12 | 12 | - | 0 | 0 | - |
| Deposition | 67 | 67 | - | 67 | 67 | - |
| Fertilization | 99 | 99 | - | 0 | 0 | - |
| Manure | 19 | 19 | - | 0 | 0 | - |
| **n**et balance (TgN yr$^{-1}$) | 44 | 45 | | 52 | 54 | |

[1]Bouwman et al. (2013),[2]Zaehle et al. (2010),[3]Braakhekke et al. (2017),[4]volatilization from natural soils (2.4 $TgN\ yr^{-1}$) from Bouwman et al. (2002),[5]volatilization from manure and synthetic fertiliser on crop- and grasslands (19 $TgN\ yr^{-1}$) from Bouwman et al. (1997),[6]Galloway et al. (2004),[7]Sutton et al. (2013),[8]Tian et al. (2019),[9]Tian et al. (2020),[10]Scheer et al. (2020),[11]Yu and Zhuang (2020)

NPP. NPP was 2.2 PgC yr$^{-1}$ lower with the *C-costly* approach compared to the *Original* approach. However, C losses from heterotrophic respiration and fire were also lower by 1.9 and 0.3 PgC yr$^{-1}$ respectively..

### 3.2.2 Dynamic land use

The simulations with dynamic land use include agricultural production and related additional N in- and outputs. Additional inputs are N from application of manure and synthetic fertilisers and additional outputs are N removed through crop harvesting, grazing and emissions from land-use change. The differences of the total BNF, soil mineral N and organic C and N stocks are similar to the PNV simulations. Between 2001 and 2010, LPJmL simulated an average N sink of 44 TgN yr$^{-1}$ for the *Original* and 45 TgN yr$^{-1}$ for the *C-costly* approach (Fig. 4 c, d, and Tab. 2). Already in 1901, the N balances of the PNV and dynamic land-use simulations diverge. Since there are no synthetic fertiliser inputs in 1901, only the relatively small additional inputs from establishment and manure were counteracted by N removal through crop harvesting and land use change emissions, which shifts the total N balance towards a smaller source. This persists even after inputs from manure and fertiliser were increased starting in the 1950s, which not only resulted in higher crop yields and therefore N removed through harvesting but

also increased N losses from emissions and leaching. As was shown for the PNV simulations, the overall N balance is similar for both approaches but with different in- and output terms driven by the higher BNF in the *Original* approach. N emissions and leaching for the *Original* approach (128 TgN yr$^{-1}$ and 74 TgN yr$^{-1}$, resp.) were higher than for the *C-costly* approach (99 TgN yr$^{-1}$ and 55 TgN yr$^{-1}$, resp.) (Tab. 2). Similarly to the PNV scenario, all types of emissions are lower with the *C-costly* approach. $NH_3$ emissions from volatilization decrease by 11 TgN yr$^{-1}$, $N_2$ emissions by 13 TgN yr$^{-1}$, $N_2O$ emissions by 3 TgN yr$^{-1}$ and fire emissions by 2 TgN yr$^{-1}$. N removal from harvesting was 142 TgN yr$^{-1}$ on average between 2001 and 2010 for the *Original* and 108 TgN yr$^{-1}$ for the *C-costly* approach. This indicates a stronger N limitation of agricultural areas in the *C-costly* approach. The majority of this reduction can be attributed to managed grassland and not croplands (Fig. S3 and 4).

Similar to the PNV simulations, the overal C balance did barely differ between the two approaches (Fig. C9 c and d) While the C input from manure and establishment was similar for both approaches, NPP was 1.8 PgC yr$^{-1}$ lower in the *C-costly* approach. C lost from land-use change was similar. Fire emissions and C removed through harvest only differed by 0.1 PgC yr$^{-1}$ while heterotrophic respiration was 1.4 PgC yr$^{-1}$ lower in the *C-costly* than in the *Original* approach.

## 4   Discussion

While the *Original* approach only indirectly accounts for temperature and water limitation, as these also limit evapotranspiration and NPP, the *C-costly* approach explicitly considers the limitation of BNF from soil temperature, water content and NPP separately. These have long been established as limiting factors for BNF. Depending on the prevailing conditions BNF may be limited more strongly by temperature or soil moisture or a combination of both. The role of temperature was explored early on by Meyer and Anderson (1959) and was followed by numerous studies for different plant species or legume crop varieties and temperature ranges (e.g. Montañez et al., 1995). Such studies allowed to quantify optimal temperature ranges and limits facilitating the development of functions such as $f_T(T_{\text{soil}})$ which are based on empirically derived temperature curves (e.g. Halliday and Pate, 1976). A similarly large literature body exists on the role of soil moisture (e.g Serraj et al., 1999; Rousk et al., 2018). Valentine et al. (2018) describe the different pathways how drought stress inhibits BNF, an important aspect being the change in nodule water potential indicating a strong connection to soil water content. While flooding of soils can also inhibit BNF through $O_2$ limitation, nitrogenase was shown to be more active in waterlogged environments (Jiang et al., 2021). Therefore, we are confident that our linear function for $f_W(\text{SWC})$ assuming that only too low soil moisture levels limit BNF (Wu and McGechan, 1999) reflects empirical observations well. As BNF is associated with a respiratory loss of C, the net amount of C assimilated via photosynthesis (NPP) available for BNF as well as the fixation efficiency (respiratory loss of C per gained N) are additional important controlling factors. A recent meta analysis by Yao et al. (2024) highlights the importance of plant taxa for BNF in addition to abiotic factors. This is in line with early experimental work that quantified respiratory loss of C per N fixed (Reed et al., 2011; Patterson and Larue, 1983; Voisin et al., 2003) and total amount of NPP spent on fixation (Kaschuk et al., 2009) for different N fixing plants and already showed that functional traits have to be considered when

assessing BNF. Therefore, including NPP as well as a cost of fixation as we did with the *C-costly* approach is an important conceptual improvement.

The *C-costly* approach is not only conceptually superior to the simplistic *Original* approach in LPJmL, it also performs better in comparison to external data. Still, some mismatches with reference data remain, such as an overestimation of BNF in the tropics (Fig 3 c). However, the ensemble mean of a recent study evaluating the N cycle of eleven DGVMs shows a similar overestimation in the tropics and a large bias indicating little agreement between models (Kou-Giesbrecht et al., 2023). They attributed this to the fact that BNF is typically modelled as a function of vegetation activity expressed either through NPP or evapotranspiration. Our results show that the overestimation of tropical BNF is reduced if temperature and water availability are considered as separate limitations, which supports their interpretation. Furthermore, the NPP that can be used for BNF depends on the overall productivity which certainly is higher in the tropics. It is likely that additional variables not considered in our approach constrain BNF there, such as phosphorus limitation (Vitousek, 1984; Lee et al., 2019). However, it has also been suggested that as a result of higher N losses, tropical BNF should be higher than observations imply (Hedin et al., 2009). This could be a result of uncertainties inherent to BNF measurements (Soper et al., 2021) or the limited amount of data available from tropical ecosystems.

Furthermore, simulated BNF was at the higher end of the range reported by Davies-Barnard and Friedlingstein (2020b) for the *C-costly* approach. One explanation is that Davies-Barnard and Friedlingstein (2020b) aggregate crop- and grassland areas assuming their BNF rates are identical. However, a recent study provides evidence that BNF of crop legumes might actually be substantially higher than that of forage legumes (Herridge et al., 2022; Peoples et al., 2021) and therefore BNF of croplands and grassland cannot be assumed to be similar. Consistent with this, we also had to select much higher potential N fixation rates for the crop PFTs compared to the other PFTs to achieve sufficient cropland BNF (Tab. 1).

We expected that limiting BNF of legume crops would result in stronger N stress and reduced yields. However, yields for the legume crops were similar between the two approaches. One explanation is the direct link of maintenance respiration of a plant organ to its N content. Reducing the N that is taken up via BNF results in a lower organ N content and maintenance respiration and thus similar NPP. Indeed, C:N ratios are higher for the *C-costly* approach compared to the *Original* approach indicating a lower plant N content (Fig. C2).

The average $\%N_{\mathrm{dfa}}$ of legume crops was between approximately 30 and 100% for the *C-costly* approach and 100% for the *Original* approach. Comparing the distribution (Fig. C7 d) to $\%N_{\mathrm{dfa}}$ observations shows that the values of the *C-costly* approach are possible but at the upper end of observations, while those of the *Original* approach are not supported by observations. For soybean, experimental values range from 0 to 98% with an average of 52% (Salvagiotti et al., 2008). Herridge et al. (2008) report average values between 40 and 75% on average and up to 97% for experiments but only 36 and 68% for farmers' fields depending on the cultivated legume crop. $\%N_{\mathrm{dfa}}$ is strongly related to soil mineral N content and thus fertilisation levels. The high $\%N_{\mathrm{dfa}}$ may be an indication that either fertiliser levels or active and passive N uptake and retranslocation of N at leaf senescence are underestimated by LPJmL and respective processes should be reevaluated. We found a higher $\%N_{\mathrm{dfa}}$ for both the natural vegetation and the cropland in warm and dry areas (Fig. C8) where mineralisation of organic N is limited (Dessureault-Rompré et al., 2010).

We expected that the differences in the BNF between the two approaches would be reflected by differences in the C stocks and fluxes due to the close link of the C and N cycles in LPJmL. Both the C inflow into terrestrial C stocks from NPP and outflows from harvest, heterotrophic respiration and fire were lower in the *C-costly* approach, leading to a similar net C balance for the two approaches (Fig. C9 and Fig. S5). Accounting for the cost of BNF in the form respiratory losses of NPP lead to lower NPP which limited biomass accumulation and in turn harvest as well as biomass available for burning and heterotrophic respiration via reduced litter accumulation. Because of the close link of the C and N cycles, the net N balance is also similar for the two approaches. The lower BNF in the *C-costly* approach results in lower N outfluxes, i.e. leaching, emissions, and harvests. The *Original* approach added mineral N to the soils of the natural vegetation and managed grassland even if the vegetation was not N limited. Legume crops that received all N they demanded as in the *Original* approach returned high N content residues to the soil, increasing N inputs and mineral N stocks. As a result, the mineral N content of soils was higher in the *Original* approach, explaining the differences in yields and leaching. Similarly, soil mineral N content influences N emissions except fire emissions, which are controlled by the N content of the burned vegetation and litter. Since this also decreased, fire emissions were lower with the *C-costly* approach. In contrast to the lower BNF, which is in line with observations, N losses from leaching and emissions (from volatilization, denitrification, nitrification, fire and land-use change) are underestimated by LPJmL simulations compared to observational data (see Tab. 2) in both approaches. The overestimation of emissions from volatilization of soil $NH_4^+$ is strongly reduced with the *C-costly* approach because the soil $NH_4^+$ pool is lower in the *C-costly* approach compared to the *Original* approach where BNF is directly added to the soil $NH_4^+$ (see sec. 2.2). While $N_2O$ emissions compare well to literature estimates, $N_2$ emissions are more strongly underestimated with the *C-costly* approach. Similarly to the soil $NH_4^+$ pool, the soil $NO_3^-$ is reduced because less $NH_4^+$ is available for nitrification resulting in reduced $N_2$ emissions. Overall, the reduction shifts emissions from an over- to an underestimation of literature values. While one source of differences is the missing representation of $NO_x$ emissions in LPJmL, this is not sufficient to fully explain the difference. However, the models of the TRENDY-N ensemble also underestimated N losses from emissions of $NH_3$, $N_2O$, $NO_x$, and $N_2$, as well as leaching (Kou-Giesbrecht et al., 2023), suggesting that processes within DGVMs and scenario assumptions need to be revised. For LPJmL, we identified several potential causes: First, the manure input accounts only for manure applied to cropland and the total amount is in line with other sources reporting cropland manure (Zhang et al., 2021) but does not account for manure added to grasslands other than the internal recycling by grazing animals (Heinke et al., 2023). Second, N losses and emissions strongly vary between different agricultural production systems whose representation would require not only the implementation of more detailed management options but also data sets on the spatial patterns of the application of different management specifics of these systems. Third, we conducted our simulations assuming cover cropping outside the growing season on all croplands, which overestimates the extent of cover cropping and reduces N losses. However, data on cover cropping systems are not available (e.g., Porwollik et al., 2022).

While the *C-costly* approach improved simulation results for BNF as well as other components of the N balance and model results are in line with other DGVMs that represent the N cycle, we see potential for further improvement. The *C-costly* approach depends on multiple parameters some of which are not well constrained. Values for the potential N fixation rate vary between species and across sites (Ma et al., 2022) and selecting one value to be representative for one PFT or even all PFTs of

an entire climate zone is a strong simplification. Furthermore, experiments have shown a large variation of the respiratory cost of BNF (Reed et al., 2011; Patterson and Larue, 1983; Voisin et al., 2003) as well as the amount of NPP different plant species invest (Kaschuk et al., 2009), which is not well reflected by the current parameterisation.

In addition, we assume a constant fraction of N fixers present in a community. However, the amount of N fixers changes over time dependent on N stress (Herben et al., 2017; Taylor et al., 2019). N fixation, the share of fixers and/or nodule abundance is low in undisturbed N-rich environments and nodules need to be produced to increase N fixation if N availability decreases (Fisher et al., 2010; Crews, 1999). Similarly, N fixation does not cease instantaneously when N becomes more abundant but is only reduced after the share of fixers and/or nodule abundance has decreased (Thornley et al., 1995; Herben et al., 2017). In contrast, fixers are always present in LPJmL and can instantly fix N if necessary. Therefore, LPJmL likely simulates too quick adaptation to changing N availability and overestimates the short term capability of the community to buffer changes in N availability.

While our approach simulates the total amount of BNF well, it does not distinguish symbiotic from free living or heterotrophic N fixation. However, these are two different sources of N and their share of total BNF shows large spatial heterogeneity (Davies-Barnard and Friedlingstein, 2020b). In contrast to symbiotic BNF, free living BNF does not require NPP expenditures and separating the two may further improve simulation results for NPP and dependent variables.

In the following we qualitatively compare our approach to common approaches used in crop models and DGVMs. A synthesis of nine crop models by Liu et al. (2011) showed that soil water status and N supply were the most widely considered control variables. Soil temperature was only considered by four and plant C supply only by two models, despite their importance for limiting BNF. The *C-costly* approach also uses empirical factors to account for soil temperature and soil water status, whereas the role of N supply, plant C supply and plant growth stage are simulated mechanistically in LPJmL, which is a clear distinction from the models assessed by Liu et al. (2011).

Our approach is at the higher end of the complexity when compared to eleven TRENDY-N DGVMs that include the N cycle. As shown in Kou-Giesbrecht et al. (2023), five DGVMs follow an approach similar to the *Original* approach calculating BNF based on evapotranspiration or NPP, three models calculate BNF as a function of N limitation, two models assume a constant BNF and in one model BNF is derived in post-processing to close the N cycle. The remaining three models use more complex approaches which can be compared to the *C-costly* approach. CLM5.0 (Lawrence et al., 2019) uses an approach based on Fisher et al. (2010), explicitly minimising the cost of active N uptake, retranslocation and BNF, and distinguishes asymbiotic from symbiotic N fixation. In CTEM, BNF is a function of temperature, vegetation cover, soil nitrate and plant structural C pools (Arain et al., 2006; Dickinson et al., 2002). The DLEM model considers soil temperature, soil moisture status, soil C and soil N (Tian et al., 2015). While The approach used in CLM5.0 is more complex compared to the *C-costly* approach and addresses some of the conceptual shortcomings of the *C-costly* approach discussed earlier, the approach used in CTEM is of similar complexity and simulates values at the upper end of recent literature estimates (Kou-Giesbrecht et al., 2023). However, global BNF values and latitudinal distribution simulated by CLM5.0 as shown by Kou-Giesbrecht et al. (2023) in Fig. 3 and A6 are comparable to those simulated with *C-costly* approach. To fully assess the advantages of such a complex approach over

the *C-costly* approach or that of CTEM or DLEM, a comparison of spatial patterns or of simulations at higher spatial resolution could be a worthwhile future endeavour.

## 5 Conclusions

Compared to the simplistic *Original* BNF implementation in LPJmL, the more complex *C-costly* approach as described here presents a substantial improvement of the representation of BNF in LPJmL. While the *Original* approach led to an overestimation of BNF and was insensitive to soil temperature and soil water conditions, the *C-costly* approach overcomes these issues and can help to better project future BNF and its effects on N limitation of the terrestrial biosphere as well as losses of reactive N to the environment, including the greenhouse gas nitrous oxide ($N_2O$). Further research is needed, especially with respect to balancing different in- and outfluxes and internal recycling rates. The current improvement of BNF simulations with LPJmL and the associated underestimation of loss terms exemplifies the scope of this problem. Our study highlights the importance of a detailed implementation of the processes controlling BNF for N cycling in DGVMs. While the *C-costly* approach already improved simulations results, we think that additional benefits would be gained by explicitly separating BNF by symbiotic and free living bacteria and from accounting for the costs of other N uptake sources except passive N uptake.

*Code availability.* The source code of LPJmL in the exact form as described here is available at zenodo.org (Wirth et al., 2023) and on https://github.com/PIK-LPJmL/LPJmL.

*Data availability.* The historical climate data from the GSWP-W5E5 dataset are available from https://doi.org/10.48364/ISIMIP.982724 (Lange et al., 2022). The historical data of atmospheric N deposition and atmospheric CO2 concentrations can be obtained from https://doi.org/10.48364/ISIMIP.600567 (Yang and Tian, 2020) and https://doi.org/10.48364/ISIMIP.664235.2 (Büchner and Reyer, 2022), respectively. All input data, model code, model outputs, and scripts that have been used to produce the results presented in this paper are archived at the Potsdam Institute for Climate Impact Research and are available upon request.

## Appendix A: Nitrogen demand and uptake

The total N demand ($N_{\text{demand}}$ in $gN\ m^{-2}$) at any time $t$ is the sum of leaf N demand for RuBisCo and structural components ($N_{\text{demand,leaf}}$ in $gN\ m^{-2}$) and the N demand for structural components of the other plant compartments.

$$N_{\text{demand,leaf}} = 25 \cdot 0.02314815/\text{daylength} \cdot V_{\max} \cdot exp(-0.02 \cdot (T - 25)) \cdot f_{\text{LAI}}(\text{LAI}) + NC_{\text{leaf,median}} \cdot C_{\text{leaf,t}}, \tag{A1}$$

where $V_{\max}$ ($gC\,m^{-2}$) is the PFT-specific maximum carboxylation capacity computed based on absorbed photosynthetically active radiation (APAR) and canopy conductance (Schaphoff et al., 2018b; Sitch et al., 2003), $T$ is the average temperature (°C) of the current day and daylength is the duration of daylight (h). $f_{\text{LAI}}(\text{LAI})$ is a dimensionless modifier to account for the

current leaf area index (von Bloh et al., 2018) and $C_{\text{leaf}}$ (gC m$^{-2}$) is the current leaf C content.

$$f_{\text{LAI}}(\text{LAI}) = \begin{cases} \max(0.1, \text{LAI}) & \text{for } \text{LAI} < 1 \\ \exp(0.08 \cdot \min(\text{LAI}, 7)) & \text{otherwise} \end{cases} \tag{A2}$$

$$C_{\text{leaf,t}} = C_{\text{leaf}} + f_{\text{leaf}} \cdot \sum_{t'=1}^{t} \text{NPP}_{t'} - \Delta\text{litter}_{t'} \tag{A3}$$

LAI is the current leaf area index and $sum_{t'=1}^{t}\text{NPP}_{t'} - \Delta\text{litter}_{t'}$ is the difference between the accumulated biomass increment and litterfall.

$$N_{\text{demand,t}} = \left(N_{\text{demand,leaf}} + \sum_{m=1}^{M} N_m + \text{NC}_t \cdot \sum_{m=1}^{M}(f_m/R_m) \cdot \sum_{t'=1}^{t}\text{NPP}_{t'} - \Delta\text{litter}_{t'}\right) \cdot (1 + k_{\text{store}}), \tag{A4}$$

where $M$ is one for grasses, two for trees and three for crops equalling the number of the respective PFT's plant compartments excluding leaves, $NC_t = \min(\max(N_{\text{leaf,t}}/C_{\text{leaf,t}}, NC_{\text{leaf,low}}), NC_{\text{leaf,high}})$, $f_m$ is the fraction of biomass allocated to the

compartment $m$, $R_m$ is C:N ratio of compartment $m$ relative to the leaf C:N ratio and $k_{\text{store}}$ is a PFT-specific parameter to maintain the PFTs' labile N storage. Passive and active N uptake ($N_{\text{uptake}}$) from each soil layer $l$ ($n_{\text{soillayer}} = 6$) is calculated as a function of the potential N uptake of the root system.

$$N_{\text{uptake}} = sum_{l=1}^{n_{\text{soillayer}}} 2 \cdot N_{\text{up,root}} \cdot C_{\text{root,t}} \cdot \text{rootdist}_l \cdot f_N(N_{\text{avail,l}}) \cdot f_T(T_{\text{soil,l}}) \cdot f_{\text{NC}}(\text{NC}_{\text{plant}}), \tag{A5}$$

where $N_{\text{up,root}}$ is the PFT-specific maximum N uptake rate per unit of fine root mass in each layer, $C_{\text{root,t}}$ is the current root

C, rootdist$_l$ is the fraction of roots in layer $l$, $f_N$, $f_T$ and $f_{\text{NC}}$ are dimensionless modifiers for the availability of mineral N, soil temperature and plant N:C ratio (von Bloh et al., 2018). $C_{\text{root,t}}$ is calculated as $C_{\text{leaf,t}}$ in Eq. A3. The root distribution can be calculated from the proportion of roots from the surface to soil depth $z$ following Jackson et al. (1996):

$$\text{rootdist}_z = \frac{1 - \beta_{\text{root}}^z}{1 - \beta_{\text{root}}^{z_{\text{bottom}}}}, \tag{A6}$$

where $z_{\text{bottom}}$ is the lower boundary of the last soil layer and $\beta_{\text{root}}$ is a PFT-specific parameter (Tab. A1). The root proportion

of one soil layer can be calculated as

$$\text{rootdist}_l = \text{rootdist}_{z(l)} - \text{rootdist}_{z(l-1)}. \tag{A7}$$

     $f_N$ follows Michaelis-Menten kinetics

$$f_N(N_{\text{avail,l}}) = k_{\text{N,min}} + \frac{N_{\text{avail,l}}}{N_{\text{avail,l}} + K_{\text{N,min}} \cdot \Theta_{\text{max}} \cdot d_{\text{soil}}}, \tag{A8}$$

where $N_{\text{avail,l}} = \text{NO}_{3,\text{soil,l}}^- + \text{NH}_{4,\text{soil,l}}^+$, $k_{\text{N,min}}$ and $K_{\text{N,min}}$ are the PFT-specific parameters describing the Michaelis-Menten

kinetics, $\Theta_{\text{max}}$ is the soil type specific fractional pore space and $d_{\text{soil,l}}$ (dimensionless) is the soil layer depth (m).

**Table A1.** PFT-specific parameters used in N demand and uptake calculations

| PFT | NC$_{\text{leaf}}$ | | | | | $R_m$ | | | $k_{\text{store}}$ | $N_{\text{up,root}}$ | $\beta_{\text{root}}$ | $k_{\text{N,min}}$ | $K_{\text{n,min}}$ |
|---|---|---|---|---|---|---|---|---|---|---|---|---|---|
| | low | median | high | root | sapwood | storage organ | pool | | | | | | |
| | - | - | - | - | - | - | - | - | gN gC$^{-1}$ d$^{-1}$ | - | - | gN m$^{-2}$ |
| TrBE | 15.6 | 26.8 | 46.2 | 1.16 | 13.5 | - | - | 0.1 | 2.8 | 0.952 | 0.05 | 1.48 |
| TrBR | 15.4 | 23.1 | 34.6 | 1.16 | 13.5 | - | - | 0.1 | 2.8 | 0.981 | 0.05 | 1.48 |
| TeNE | 31.8 | 45.0 | 63.8 | 1.16 | 13.5 | - | - | 0.1 | 2.8 | 0.976 | 0.05 | 1.48 |
| TeBE | 15.6 | 26.8 | 46.2 | 1.16 | 13.5 | - | - | 0.1 | 2.8 | 0.964 | 0.05 | 1.48 |
| TeBS | 15.4 | 23.1 | 34.6 | 1.16 | 13.5 | - | - | 0.1 | 2.8 | 0.966 | 0.05 | 1.48 |
| BoNE | 31.8 | 45.0 | 63.8 | 1.16 | 13.5 | - | - | 0.1 | 2.8 | 0.955 | 0.05 | 1.48 |
| BoBS | 15.4 | 23.1 | 34.6 | 1.16 | 13.5 | - | - | 0.1 | 2.8 | 0.955 | 0.05 | 1.48 |
| BoNS | 18.4 | 26.0 | 36.9 | 1.16 | 13.5 | - | - | 0.1 | 2.8 | 0.955 | 0.05 | 1.48 |
| TrH | 17.4 | 34.0 | 66.9 | 1.16 | - | - | - | 0.05 | 5.51 | 0.973 | 0.05 | 1.19 |
| TeH | 10.5 | 19.9 | 37.9 | 1.16 | - | - | - | 0.05 | 5.51 | 0.943 | 0.05 | 1.19 |
| PoH | 10.5 | 19.9 | 37.9 | 1.16 | - | - | - | 0.05 | 5.51 | 0.943 | 0.05 | 1.19 |
| Soybean | 14.3 | 25.0 | 58.8 | 1.16 | - | 0.42 | 3 | 0.1 | 5.51 | 0.969 | 0.05 | 1.48 |
| Pulses | 14.3 | 25.0 | 58.8 | 1.16 | - | 0.42 | 3 | 0.1 | 5.51 | 0.969 | 0.05 | 1.48 |

$f_T$ is the temperature function given by Thornley (1991):

$$f_T(T_{\text{soil,l}}) = \max\left(\frac{(T_{\text{soil,l}} - T_0) \cdot (2 \cdot T_m - T_0 - T_{\text{soil,l}})}{(T_r - T_0) \cdot (2 \cdot T_m - T_0 - T_r)}, 0\right), \tag{A9}$$

where $T_0 < T_r < 2 \cdot T_m - T_0$ has to be fulfilled. von Bloh et al. (2018) defined $T_m = 15°C$, $T_r = 15°C$ and $T_0 = -25°C$ which leads to the maximum of one at temperatures of 15°C and higher and non-zero values above -25°C.

$f_{\text{NC}}$ was taken from Zaehle and Friend (2010):

$$f_{\text{NC}} = \min\left(\max\left(\frac{\text{NC}_{\text{plant}} - \text{NC}_{\text{leaf,high}}}{\text{NC}_{\text{leaf,low}} - \text{NC}_{\text{leaf,high}}}, 0\right), 1\right), \tag{A10}$$

where $\text{NC}_{\text{plant}} = \frac{N_{\text{leaf}} + N_{\text{root}}}{C_{\text{leaf}} + C_{\text{root}}}$, $\text{NC}_{\text{leaf,min}}$ and $\text{NC}_{\text{leaf,max}}$ are PFT-specific parameters extracted from the TRY database (Kattge et al., 2020) (Tab. A1).

The labile N $N_{\text{labile,t}}$ are the current reserves which have accumulated via N uptake and retranslocation.

$$N_{\text{labile,t}} = N_{\text{labile}} + \sum_{t'=1}^{t} N_{\text{uptake,t'}} - N_{\text{resorb,t'}} \tag{A11}$$

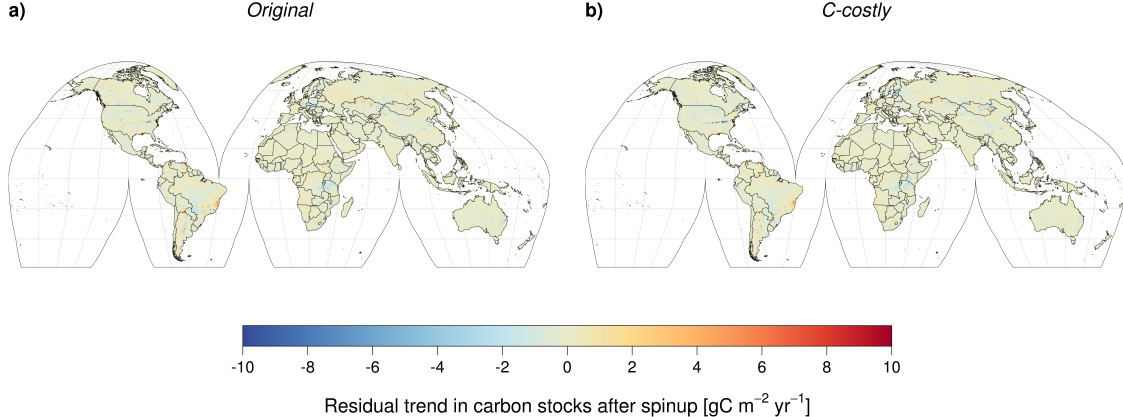

**Figure B1.** Residual trends in C stocks after the spinup simulation averaged over 1000 years for the Original (a) and the C-costly (b) approach.

## Appendix B:  Spinup simulation carbon stocks

With constant forcing (i.e., stable pre-industrial atmospheric $CO_2$ concentration, and atmospheric N deposition, and climate), the global C stocks showed a residual trend of -0.0106 $\mathrm{PgC\ yr^{-1}}$ for the *Original* approach and -0.0121 $\mathrm{PgC\ yr^{-1}}$ for the *C-costly* approach. This is 8-10 times lower than the steady-state criterion of 0.1 $\mathrm{PgC\ yr^{-1}}$ residual trend after spinup, which is used by the Global Carbon Project to validate DGVMs for inclusion in their global C budget analysis (Friedlingstein et al., 2022). At the grid cell level, the vast majority of cells (94 % for the *Original* approach and 95 % for the *C-costly* approach) exhibited residual trends in total C stocks of less than $\pm\ 1\ \mathrm{gC\ m^2\ yr^{-1}}$. The corresponding maps are shown in Fig. B1.

## Appendix C:  Additional figures and tables

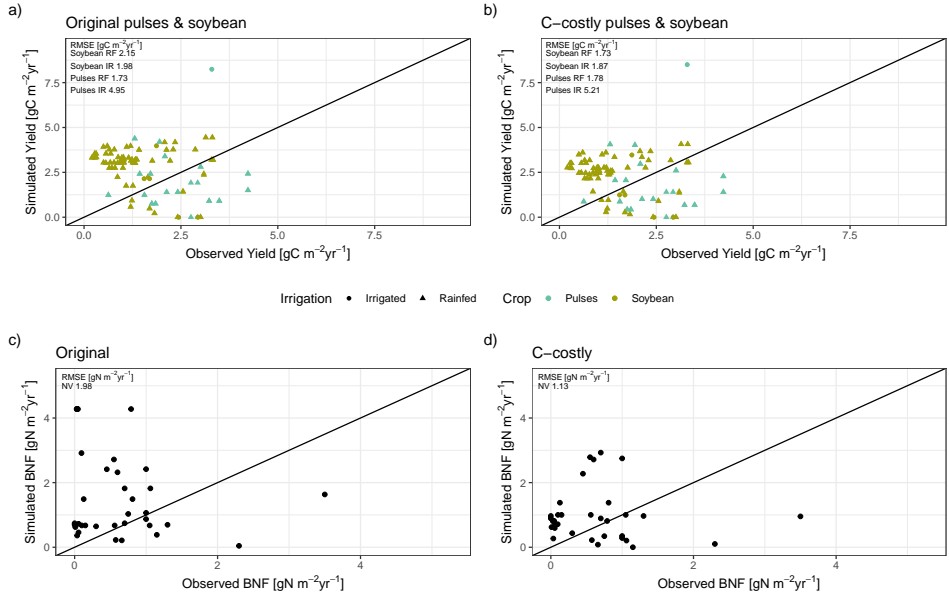

**Figure C1.** Simulated and observed crop yields (a,b) for soybean (green) and pulses (blue) and BNF in natural vegetation (c,d).

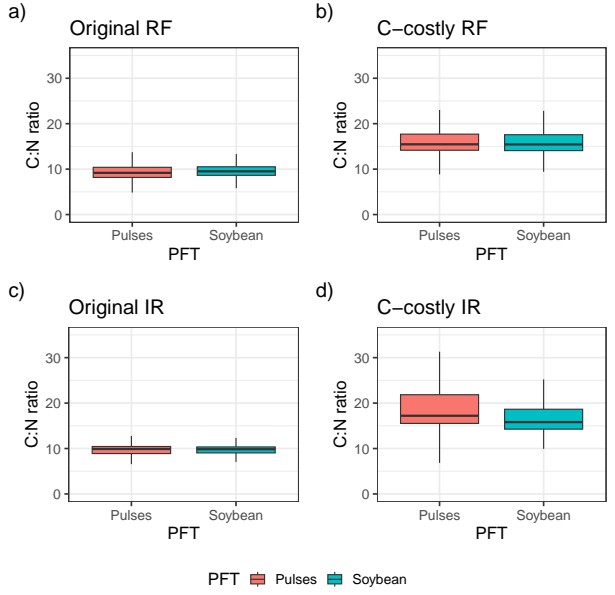

**Figure C2.** Vegetation C:N ratio for years 2001 to 2010 for rainfed (RF) and irrigated (IR) soybean (red) and pulses (blue) for the *Original* and *C-costly* approach.

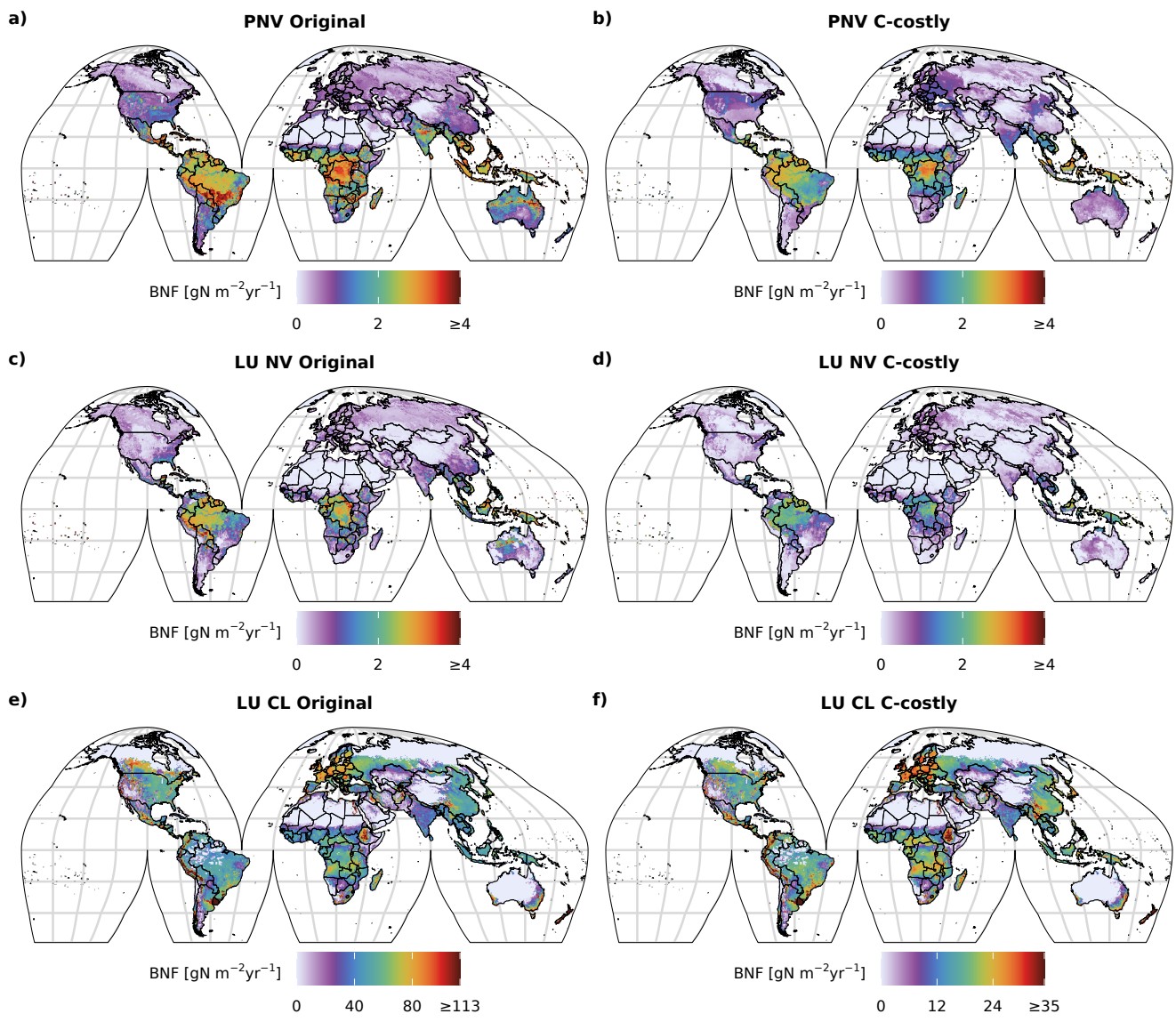

**Figure C3.** 2001 to 2010 average BNF in $\text{gNm}^{-2}yr^{-1}$ of the potential natural vegetation simulations (PNV) (a,b) and of the natural vegetation (NV) (c,d) and managed land (AG) (e,f) area fractions of the dynamic land-use (LU) simulations.

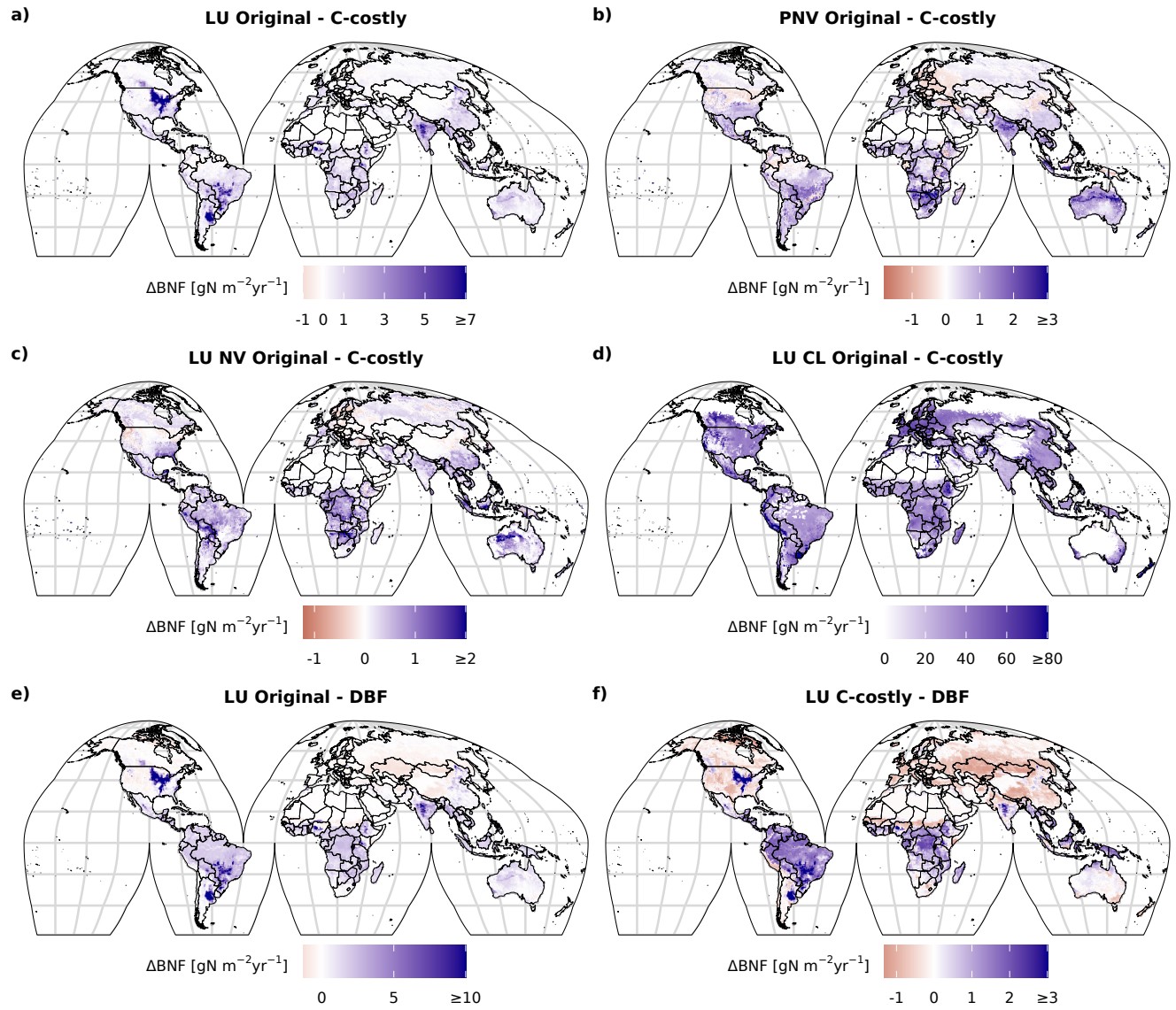

**Figure C4.** Difference between 2001 to 2010 average BNF in $gNm^{-2}yr^{-1}$ between the two approaches (a-d) for the dynamic land-use (LU) simulations (a), the potential natural vegetation simulations (PNV) (b), for the area fractions of natural vegetation (NV) (c) and managed land (AG) (e) of the dynamic land-use simulations and difference to the data from Davis-Barnard & Friedlingstein 2002 (DBF) (e,f).

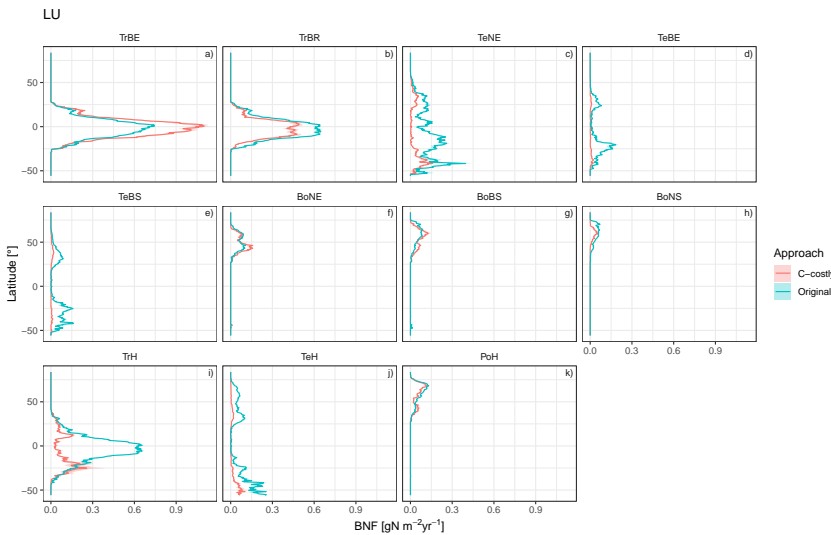

**Figure C5.** Latitudinal distribution of BNF for each PFT for the dynamic land-use simulations for the *Original* (red) and *C-costly* approach (blue).

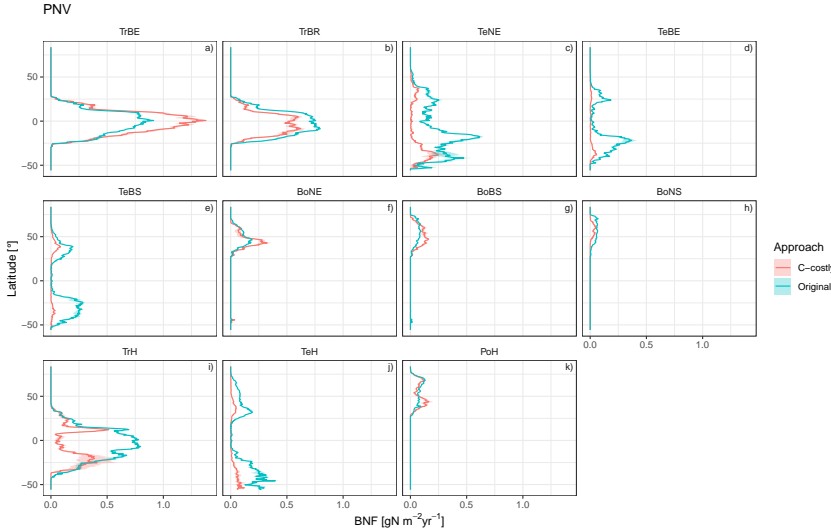

**Figure C6.** Latitudinal distribution of BNF for each PFT for the potential natural vegetation simulations for the *Original* (red) and *C-costly* approach (blue).

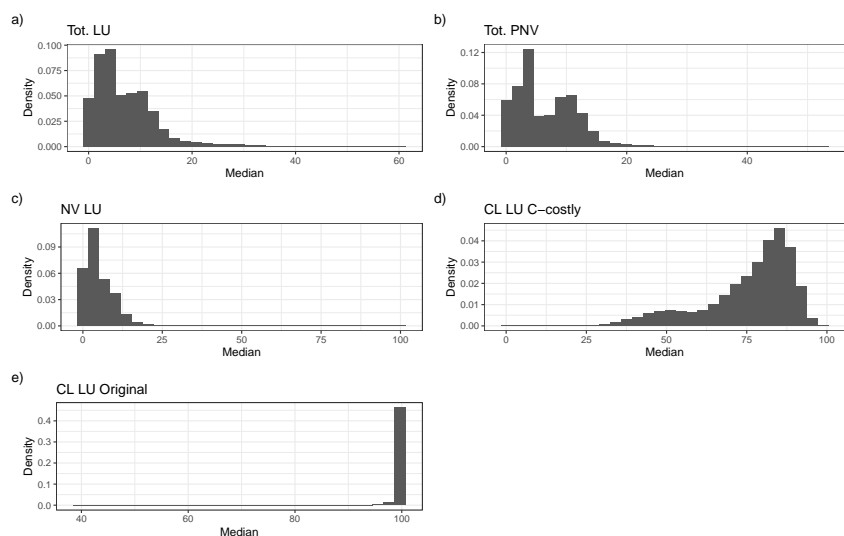

**Figure C7.** Density distribution of the fraction of BNF of total N uptake for the dynamic land-use simulations (a), potential natural vegetation (b) and for the area fractions of natural vegetation (NV) (c) and cropland (CL) using the *C-costly* (d) and the *Original* (e) approach for the dynamic land-use simulations.

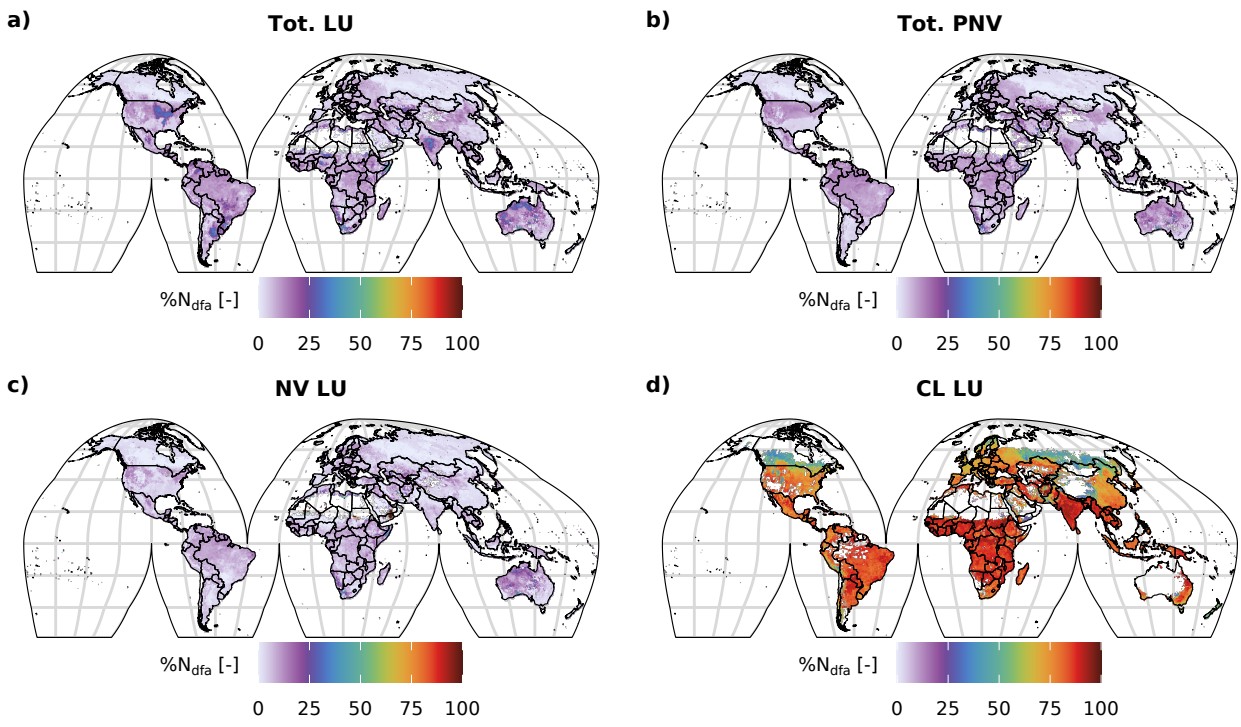

**Figure C8.** Global distribution of the fraction of $\%N_{dfa}$ for the dynamic land-use (a) and potential natural vegetation simulations (b) and the natural vegetation (c) and cropland (d) fraction of the dynamic land-use simulation.

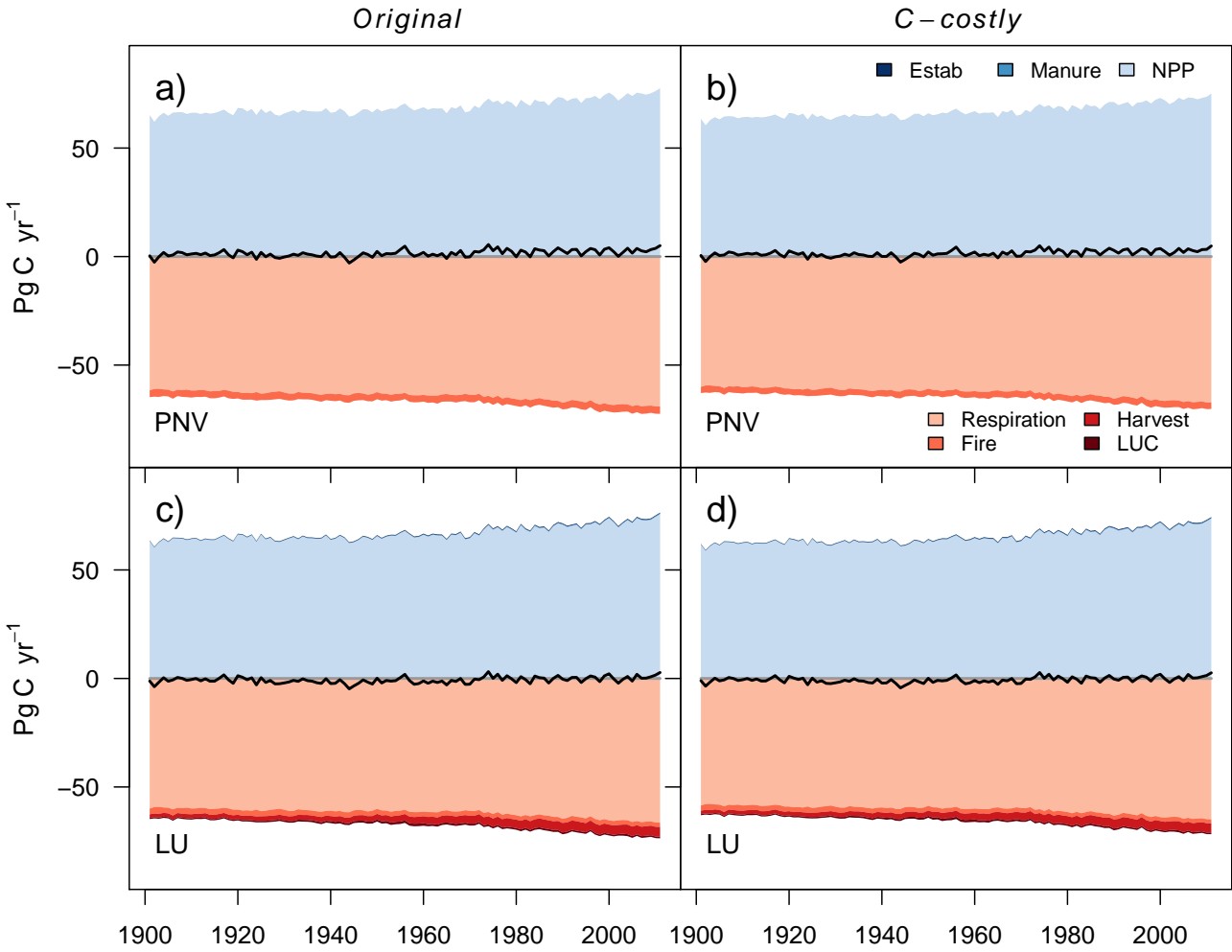

**Figure C9.** Global terrestrial C balance. Scenarios include the *Original* approach, the *C-costly* approach for natural vegetation and actual land use. Net balance is denoted by the black line. C inputs include C from manure, PFT establishment (Estab) and NPP. C losses include heterotrophic respiration, fire emissions, harvested C and land-use change emissions (from deforestation and product turnover).

*Author contributions.* SBW, CM, FS, SR, Ssch and WvB designed the study. SBW designed and conducted the model implementation with inputs from CM, JB, SR, Ssch and WvB. SBW, CM, FS, SR, Ssch, WvB, SO, JH and JB contributed to general model development and evaluation. SBW conducted the model simulations and wrote the original draft of the manuscript with inputs from CM, FS, SR, Ssch and WvB. SBW, CM, FS, SR, Ssch, WvB, SO, JH and JB reviewed and edited the original draft. All authors discussed the simulation results and reviewed and edited the manuscript.

*Competing interests.* At least one of the (co-)authors is a member of the editorial board of Geoscientific model development.

*Acknowledgements.* SBW acknowledges financial support from the Evangelisches Studienwerk Villigst foundation, under the research program: "Third Ways of Feeding The World" and the German Federal Ministry for Education and Research (BMBF) within the projects EXIMO (grant no. 01LP1903D) and ABCDR (grant no. 01LS2105A). JB acknowledges funding from the European Union's Horizon 2020 research and innovation programme (grant agreement No 869192). JH acknowledges funding from European Commissions's Horizon 2020 project ESM2025 – Earth System Models for the Future (grant no. 101003536). SR acknowledges financial support by the German Federal Ministry for Education and Research (BMBF) within the project ABCDR (grant no. 01LS2105A). FS is funded by the Global Challenges Foundation via Future Earth. SO acknowledges funding by the Conservation International Foundation (grant Number CI-114129).

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
