# Peer review of "Biological nitrogen fixation of natural and agricultural vegetation simulated with LPJmL 5.7.9"

_EGUsphere, 2023_

## Author Comment (AC1)

Review #1

This manuscript describes an effort to make the representation of BNF more realistic in LPJmL. The main results are that changing the representation of BNF – from a less-mechanistic function of AET to a more-mechanistic dependence on temperature, moisture, and N limitation – decreases overall estimates of BNF and modifies the spatial distribution, resulting in a better overall fit to data. This is a worthwhile effort, and from what I can tell the work is solid. My hope is that my feedback below will improve the work.

We cordially thank the reviewer for their positive evaluation of our manuscript and the constructive feedback. Below we provide a point by point response to their feedback and suggested changes to improve the manuscript.

Major/overall suggestions:

My main areas of feedback are (1) a greater focus on relevant empirical work, (2) greater discussion of how the implementation of BNF compares to other models, and (3) more explanation of the methods.

(1) I understand that this is a modeling study, but there is a lot of relevant empirical literature that is not referenced. For example, the parameters describing responses to moisture and temperature are taken from Yu & Zhuang, which is another modeling study. That's fine, but I would like to see more explanation of how those values compare to actual measurements of these quantities. As another example, your BNFfrac (see below as well) is a commonly measured quantity in N fixation work at the plant scale. Particularly for agricultural systems, there are large amounts of data. How do your results compare to empirical data? There are a few papers cited in the discussion about how N fixation varies as a function of N limitation, succession, etc., but there is a lot of work in these areas, and the discussion reads as if these were the first few that came up in a search rather than a synthesis of deep reading on the subject.

We agree that an extended comparison to empirical literature will strengthen the discussion and improve the overall quality of the manuscript. We propose to incorporate the findings of the articles listed in the table below. If possible, we selected the studies according to two criteria to limit the scope: First, we included recent meta analyses and reviews to capture the extent of available literature and, secondly, we refer to early experimental work to highlight how well this knowledge is already established. We trust that these will provide sufficient empirical context for a model development study.

| Process | Literature |
|---|---|
| Temperature limitation | (Meyer and Anderson, 1959; Montañez et al., 1995; Rousk et al., 2018; Yao et al., 2024) |
| Water limitation | (Rousk et al., 2018; Serraj et al., 1999; Valentine et al., 2018; Yao et al., 2024) |
| Carbon cost and NPP limitation | (Kaschuk et al., 2009; Patterson and Larue, 1983; Ryle et al., 1979; Voisin et al., 2003) |
| $BNF_{frac}$ or $\%N_{dfa}$ | (Herridge et al., 2008; Salvagiotti et al., 2008) |

(2) The discussion of other model implementations of more mechanistic BNF could also be improved. Ma et al. (another version of LPJ) and Yu & Zhuang (TEM) are referenced most heavily, and Fisher et al. and Davies-Barnard & Friedlingstein are also mentioned, but there are many other implementations out there ranging from land models to ecosystem models. Kou-Giesbrecht et al. 2023 (cited in the ms) has a nice table that lists a few of the TRENDY models that incorporate mechanistic representations of BNF: CLASSIC, CLM, DLEM, OCN. There are other non-TRENDY models that have been developed that have been applied at large spatial scales – LM3/LM4 and QUINCY come to mind – and there are tons of ecosystem models (ED, MEL, CENTURY, etc.) that do something similar. Readers will want to know how your implementation compares.

Thank you for this comment. We agree that this will improve the discussion and propose to conceptually compare our approach against a selection of the approaches synthesized in Kou-Giesbrecht et al., 2023 and Liu et al., 2011.

(3) Explanation of the methods: I'd like to see a clearer description in the methods of how the versions were evaluated. I'd also like to see more detail about how N limitation is calculated, given that this is the key aspect of the paper. In particular, how is Ndeficit calculated?

We propose to rename the section evaluation data to model evaluation and extend it by a description of the equations used for the global and site specific evaluation. We further propose to update the modelling protocol to thoroughly describe the simulations conducted for the site-specific evaluation.

To provide a better picture on the N limitation we will add an overview of the N demand, uptake and stress components as described in von Bloh et al. (2018) including the main equations. We will also summarize how the N deficit is calculated based on N demand and passive and active N uptake.

Minor suggestions:

Given that you've stated that you're modeling all BNF, not just legume-associated BNF, I suggest changing the name of $f_{legume}$ to $f_{fixer}$ or something like that.

Thank you for this suggestion which we will adopt.

Fig. 3 caption needs to specify what "DBF" is. I assume Davies-Barnard & Friedlingstein, but it would be nice to see in the caption, particularly given that there are other meanings of DBF (e.g., deciduous broadleaf forest).

We will include the explanation for DBF which is indeed Davies-Barnard & Friedlingstein in the caption.

The color scales on the global figures overemphasize the high range, making it hard to see variation in the lower range. For example, Fig. 3 looks largely like a map of agricultural BNF.

We agree that the color scale can be improved and propose to use the smooth rainbow scale from the khroma package (Frerebeau et al., 2024) that is able to show difference in the higher and lower part of the range.

189: In the empirical literature, what you describe as $BNF_{frac}$ is called $\%N_{dfa}$ (percent of N derived from fixation activity or derived from the atmosphere, depending on who you ask). It might help your paper to make the connection.

Thank you for establishing the connection. We will adopt the term and include the variable in our evaluation (see also response to major comment 2).

197: It's true that 4 g N/m$^2$/yr is a lot lower than 15, but 4 g N/m$^2$/yr is still a huge difference.

We will rephrase this to highlight that 4 gN/m$^2$ is still substantial.

**Bibliography**

Frerebeau, N., Lebrun, B., Arel-Bundock, V., Stervbo, U., 2024. khroma: Colour Schemes for Scientific Data Visualization.

Herridge, D.F., Peoples, M.B., Boddey, R.M., 2008. Global inputs of biological nitrogen fixation in agricultural systems. Plant Soil 311, 1–18. https://doi.org/10.1007/s11104-008-9668-3

Kaschuk, G., Kuyper, T.W., Leffelaar, P.A., Hungria, M., Giller, K.E., 2009. Are the rates of photosynthesis stimulated by the carbon sink strength of rhizobial and arbuscular mycorrhizal symbioses? Soil Biology and Biochemistry 41, 1233–1244. https://doi.org/10.1016/j.soilbio.2009.03.005

Kou-Giesbrecht, S., Arora, V.K., Seiler, C., Arneth, A., Falk, S., Jain, A.K., Joos, F., Kennedy, D., Knauer, J., Sitch, S., O'Sullivan, M., Pan, N., Sun, Q., Tian, H., Vuichard, N., Zaehle, S., 2023. Evaluating nitrogen cycling in terrestrial biosphere models: a disconnect between the carbon and nitrogen cycles. Earth Syst. Dynam. 14, 767–795. https://doi.org/10.5194/esd-14-767-2023

Liu, Y., Wu, L., Baddeley, J.A., Watson, C.A., 2011. Models of biological nitrogen fixation of legumes. A review. Agronomy Sust. Developm. 31, 155–172. https://doi.org/10.1051/agro/2010008

Meyer, D.R., Anderson, A.J., 1959. Temperature and Symbiotic Nitrogen Fixation. Nature 183, 61–61. https://doi.org/10.1038/183061a0

Montañez, A., Danso, S.K.A., Hardarson, G., 1995. The effect of temperature on nodulation and nitrogen fixation by five *Bradyrhizobium japonicum* strains. Applied Soil Ecology 2, 165–174. https://doi.org/10.1016/0929-1393(95)00052-M

Patterson, T.G., Larue, T.A., 1983. Root Respiration Associated with Nitrogenase Activity (C2H2) of Soybean, and a Comparison of Estimates 1. Plant Physiology 72, 701–705. https://doi.org/10.1104/pp.72.3.701

Rousk, K., Sorensen, P.L., Michelsen, A., 2018. What drives biological nitrogen fixation in high arctic tundra: Moisture or temperature? Ecosphere 9, e02117. https://doi.org/10.1002/ecs2.2117

Ryle, G.J.A., Powell, C.E., Gordon, A.J., 1979. The Respiratory Costs of Nitrogen Fixation in Soyabean, Cowpea, and White Clover: I. Nitrogen fixation and the respiration of the

nodulated root. Journal of Experimental Botany 30, 135–144. https://doi.org/10.1093/jxb/30.1.135

Salvagiotti, F., Cassman, K.G., Specht, J.E., Walters, D.T., Weiss, A., Dobermann, A., 2008. Nitrogen uptake, fixation and response to fertilizer N in soybeans: A review. Field Crops Research 108, 1–13. https://doi.org/10.1016/j.fcr.2008.03.001

Serraj, R., Sinclair, T.R., Purcell, L.C., 1999. Symbiotic N2 fixation response to drought. Journal of Experimental Botany 50, 143–155. https://doi.org/10.1093/jxb/50.331.143

Valentine, A.J., Benedito, V.A., Kang, Y., 2018. Legume Nitrogen Fixation and Soil Abiotic Stress: From Physiology to Genomics and Beyond, in: Annual Plant Reviews Online. John Wiley & Sons, Ltd, pp. 207–248. https://doi.org/10.1002/9781119312994.apr0456

Voisin, A.S., Salon, C., Jeudy, C., Warembourg, F.R., 2003. Symbiotic N2 fixation activity in relation to C economy of Pisum sativum L. as a function of plant phenology. Journal of Experimental Botany 54, 2733–2744. https://doi.org/10.1093/jxb/erg290

von Bloh, W., Schaphoff, S., Müller, C., Rolinski, S., Waha, K., Zaehle, S., 2018. Implementing the Nitrogen cycle into the dynamic global vegetation, hydrology and crop growth model LPJmL (version 5). Geoscientific Model Development.

Yao, Y., Han, B., Dong, X., Zhong, Y., Niu, S., Chen, X., Li, Z., 2024. Disentangling the variability of symbiotic nitrogen fixation rate and the controlling factors. Global Change Biology 30, e17206. https://doi.org/10.1111/gcb.17206

---

## Author Comment (AC2)

Review #2

The paper present a new parameterization of biological nitrogen fixation in the LPJmL model. This new parameterization, compared to the original one, takes into account the nitrogen limitation and a carbon cost for acquisition of the BNF. This is a very important improvement as it means that nitrogen fixation is directly linked to the biological activity and nitrogen limitation, which was not the case before. Hence, the total BNF fixation is reduced compared to the original formulation, which is more in agreement with observations. So it is an important improvement for LPJmL. The paper is sound and well written. So I have only few remarks, only some suggestions to improve the paper:

Thank you very much for the positive evaluation of our paper. We highly appreciate the feedback and give a point by point response with our suggested changes below.

Even if the original approach of simulation of BNF fixation has already been published, it would help the reader to present the original equations and then to show in detail what are the difference between "original" and "C-costly" parameterization. For instance, we understand only in the discussion about BNF fraction that the 2 parameterization are different not only on the calculation of N but also on the way this N is taken by the plant, directly in the new parameterization and mixed with soil mineral N in original which is also an important difference. Then it is important to give more details about the parameterization and how they differ. More generally, it would be also interesting to compare the new parameterization to parameterization used in others DGVMS that implements BNF.

We agree that this will facilitate the comparison of the two approaches and will include a description of the original approach. We will also include a conceptual comparison to BNF approaches of the models synthesized in Kou-Giesbrecht et al., 2023 and Liu et al., 2011 as also explained in our response to major comment three of reviewer one.

The results focus only on BNF, but it would be interesting to see also at global scale what is the impact of the new parameterization on the carbon cycle (for instance impact on NPP, NBP). Only the impact on legumes yield is shown if figure B1.

As the carbon and nitrogen cycle are closely linked, the overall change in NBP will qualitatively be similar to that of the overall N balance (see Fig. S5). However, we agree that explicitly showing the main C balance components will increase the informative value of the paper and will add a figure in the style of Fig. 4 for the C balance.

Also on figure 4 we see the relatively large impact of the new BNF parameterization on N emissions. It would then be interesting to show a comparison of these simulated fluxes to observations, as it is done for BNF. Especially for N2O emissions. It is obviously an important component of the GHG budget. So, with the new BNF parameterization, does it improve the simulated fluxes of N2O ?

We included a comparison to global literature estimates of N emissions in Tab. B1. We propose to move it into the main text and extend the respective sections of results and the

discussion. We will also include additional literature on N2O emissions in the discussion (e.g. Scheer et al., 2020).

Minor remarks:

l 268: The authors seem surprised that the new approach does not limit the crop yield. But if I understood well the model, it is not so surprising for me. Since in condition when NPP is not a limiting factor for BNF (that should be the case for crops) and, as the model try to fulfill the N limitation, then the simulated BNF should be sufficient to fill the N demand of the plant and then should not produce N limitation ? Then it could explain why even if the different approaches give different BNF there is no impact of yield. This is exactly what we expect from the new formulation compared to original: define the BNF to avoid N limitation but without N excess... This is also the reason why it would be interesting to show the global impact on NPP: We should expect a decrease in NPP on carbon and N limited ecosystems, as the C-cost or N stress could be too high to be fulfilled by the BNF fixation. On the contrary, we should have no change in ecosystem with few limitations even is the BNF is reduced.

Thank you for bringing this up. While NPP may not be limiting for BNF, we expected it to be lower because of the investment cost for BNF which is subtracted from the NPP, thereby reducing NPP available for plant growth and grain formation. We therefore expected a reduction in legume crop yield compared to the implementation in which the crops got all the N they need for free. We show that global yields of soybean and pulses are reduced (L248, Fig S3 and S4). However, we expected a stronger reduction. We provided one explanation which is that the reduced respiratory losses of NPP balance expenses for BNF (L269-273 and Fig. B2).

Nevertheless, we agree that including the additional insights from the C balance assessment (major comment two) will further improve the discussion of this aspect.

Figure 2: what are the percentage indicated in blue and red in a) ?

These indicate the overlap between simulated and observed ranges. We will add the explanation to the caption and provide the formula in the evaluation section.

Figure 3: the DBf term is not defined. I guess it is the observation, but it should be described

Thank you. This was also pointed out by reviewer 1 and is indeed referring to the Davies-Barndard and Friedlingstein data. We agree that it needs to be explained in the caption.

**Bibliography**

Kou-Giesbrecht, S., Arora, V.K., Seiler, C., Arneth, A., Falk, S., Jain, A.K., Joos, F., Kennedy, D., Knauer, J., Sitch, S., O'Sullivan, M., Pan, N., Sun, Q., Tian, H., Vuichard, N., Zaehle, S., 2023. Evaluating nitrogen cycling in terrestrial biosphere models: a disconnect between the carbon and nitrogen cycles. Earth Syst. Dynam. 14, 767–795. https://doi.org/10.5194/esd-14-767-2023

Liu, Y., Wu, L., Baddeley, J.A., Watson, C.A., 2011. Models of biological nitrogen fixation of legumes. A review. Agronomy Sust. Developm. 31, 155–172. https://doi.org/10.1051/agro/2010008

Scheer, C., Fuchs, K., Pelster, D.E., Butterbach-Bahl, K., 2020. Estimating global terrestrial denitrification from measured N2O:(N2O + N2) product ratios. Current Opinion in Environmental Sustainability, Climate Change, Reactive Nitrogen, Food Security and Sustainable Agriculture 47, 72–80. https://doi.org/10.1016/j.cosust.2020.07.005

---

## Author Response (AR1)

Review #1

This manuscript describes an effort to make the representation of BNF more realistic in LPJmL. The main results are that changing the representation of BNF – from a less-mechanistic function of AET to a more-mechanistic dependence on temperature, moisture, and N limitation – decreases overall estimates of BNF and modifies the spatial distribution, resulting in a better overall fit to data. This is a worthwhile effort, and from what I can tell the work is solid. My hope is that my feedback below will improve the work.

We cordially thank the reviewer for their positive evaluation of our manuscript and the constructive feedback. Below we provide a point by point response to their feedback and suggested changes to improve the manuscript.

Major/overall suggestions:

My main areas of feedback are (1) a greater focus on relevant empirical work, (2) greater discussion of how the implementation of BNF compares to other models, and (3) more explanation of the methods.

(1) I understand that this is a modeling study, but there is a lot of relevant empirical literature that is not referenced. For example, the parameters describing responses to moisture and temperature are taken from Yu & Zhuang, which is another modeling study. That's fine, but I would like to see more explanation of how those values compare to actual measurements of these quantities. As another example, your BNFfrac (see below as well) is a commonly measured quantity in N fixation work at the plant scale. Particularly for agricultural systems, there are large amounts of data. How do your results compare to empirical data? There are a few papers cited in the discussion about how N fixation varies as a function of N limitation, succession, etc., but there is a lot of work in these areas, and the discussion reads as if these were the first few that came up in a search rather than a synthesis of deep reading on the subject.

We agree that an extended comparison to empirical literature strengthens the discussion and improves the overall quality of the manuscript. We included a section at the beginning of the discussion (L301-321) relating our approach to the empirical literature on limiting factors for BNF listed in the table below. We selected studies according to two criteria to limit the scope: First, we included recent meta analyses and reviews to capture the extent of available literature and, secondly, we refer to early experimental work to highlight how well this knowledge is already established. We further extended the discussion on the simulated %Ndfa and compared the estimates for both our approaches to empirical data (L345-352).

| Process | Literature used |
| --- | --- |
| Temperature limitation | (Halliday and Pate, 1976; Meyer and Anderson, 1959; Montañez et al., 1995) |
| Water limitation | (Jiang et al., 2021; Rousk et al., 2018; Serraj et al., 1999; Valentine et al., 2018; Wu and McGechan, 1999) |
| Carbon cost and NPP limitation | (Kaschuk et al., 2009; Patterson and Larue, 1983; Reed et al., 2011; Voisin et al., 2003; Yao et al., 2024) |
| $BNF_{frac}$ or $\%N_{dfa}$ | (Herridge et al., 2008; Salvagiotti et al., 2008) |

(2) The discussion of other model implementations of more mechanistic BNF could also be improved. Ma et al. (another version of LPJ) and Yu & Zhuang (TEM) are referenced most heavily, and Fisher et al. and Davies-Barnard & Friedlingstein are also mentioned, but there are many other implementations out there ranging from land models to ecosystem models. Kou-Giesbrecht et al. 2023 (cited in the ms) has a nice table that lists a few of the TRENDY models that incorporate mechanistic representations of BNF: CLASSIC, CLM, DLEM, OCN. There are other non-TRENDY models that have been developed that have been applied at large spatial scales – LM3/LM4 and QUINCY come to mind – and there are tons of ecosystem models (ED, MEL, CENTURY, etc.) that do something similar. Readers will want to know how your implementation compares.

Thank you for this comment. We agree that it will improve the discussion to conceptually compare our approach against a selection of the approaches synthesized in Kou-Giesbrecht et al., 2023 and Liu et al., 2011.

We included a section in the discussion in which we qualitatively compare our approach against the empirical approaches typically used in crop models as reviewed by Liu et al. (2011) and the more complex DGVM approaches of CLM5.0, CLASSIC and DLEM (L405-425).

(3) Explanation of the methods: I'd like to see a clearer description in the methods of how the versions were evaluated. I'd also like to see more detail about how N limitation is calculated, given that this is the key aspect of the paper. In particular, how is Ndeficit calculated?

We renamed the section "Evaluation data" to "Model evaluation" and included equations for the calculation of the global BNF, the overlap between literature estimates and simulation results and the root mean square error (RMSE) (L160-169 and 173-176). For greater clarity, we included a sentence stating that the additional simulations conducted for the evaluation of legume crop yields and BNF follow the same protocol as the global simulations (L171ff)

To provide a better picture on the N limitation we added a subsection "BNF-relevant nitrogen cycle components in LPJmL", in which we qualitatively describe how the N deficit is determined (L101-114) and a section in the appendix in which we provide additional details including the underpinning equations and PFT-specific parameters (L444-485).

Minor suggestions:

Given that you've stated that you're modeling all BNF, not just legume-associated BNF, I suggest changing the name of $f_{legume}$ to $f_{fixer}$ or something like that.

Thank you for this suggestion which we changed throughout the manuscript.

Fig. 3 caption needs to specify what "DBF" is. I assume Davies-Barnard & Friedlingstein, but it would be nice to see in the caption, particularly given that there are other meanings of DBF (e.g., deciduous broadleaf forest).

We included the explanation for DBF, which is indeed Davies-Barnard & Friedlingstein, in the caption.

The color scales on the global figures overemphasize the high range, making it hard to see variation in the lower range. For example, Fig. 3 looks largely like a map of agricultural BNF.

We agree that the color scale can be improved and propose to use the smooth rainbow scale from the khroma package (Frerebeau et al., 2024) that is able to show difference in the higher and lower part of the range.

We updated the global figures to use the smooth rainbow scale from the khroma package (Frerebeau et al., 2024) and now only show differences for values below the 99th percentile to avoid that outlier values make major patterns hard to see.

189: In the empirical literature, what you describe as $BNF_{frac}$ is called $\%N_{dfa}$ (percent of N derived from fixation activity or derived from the atmosphere, depending on who you ask). It might help your paper to make the connection.

Thank you for establishing the connection.

We replaced the term $BNF_{frac}$ with $\%N_{dfa}$ throughout the manuscript.

197: It's true that 4 g N/m$^2$/yr is a lot lower than 15, but 4 g N/m$^2$/yr is still a huge difference.

We removed the word "only" from the sentence (L235) as we did not want to imply that this is negligible.

**Review #2**

The paper present a new parameterization of biological nitrogen fixation in the LPJmL model. This new parameterization, compared to the original one, takes into account the nitrogen limitation and a carbon cost for acquisition of the BNF. This is a very important improvement as it means that nitrogen fixation is directly linked to the biological activity and nitrogen limitation, which was not the case before. Hence, the total BNF fixation is reduced compared to the original formulation, which is more in agreement with observations. So it is an important improvement for LPJmL. The paper is sound and well written. So I have only few remarks, only some suggestions to improve the paper:

Thank you very much for the positive evaluation of our paper. We highly appreciate the feedback and give a point by point response with our suggested changes below.

Even if the original approach of simulation of BNF fixation has already been published, it would help the reader to present the original equations and then to show in detail what are the difference between "original" and "C-costly" parameterization. For instance, we understand only in the discussion about BNF fraction that the 2 parameterization are different not only on the calculation of N but also on the way this N is taken by the plant, directly in the new parameterization and mixed with soil mineral N in original which is also an important difference. Then it is important to give more details about the parameterization and how they differ. More generally, it would be also interesting to compare the new parameterization to parameterization used in others DGVMS that implements BNF.

We agree that this will facilitate the comparison of the two approaches and now include a brief description of the original approach in the newly added section BNF-relevant nitrogen cycle components (L115-119) and a qualitative comparison to the approaches used by other DGVMs that were synthesized in Kou-Giesbrecht et al., 2023 and Liu et al., 2011 (L405-425) in the discussion. See also our response to major comment three of reviewer one.

The results focus only on BNF, but it would be interesting to see also at global scale what is the impact of the new parameterization on the carbon cycle (for instance impact on NPP, NBP). Only the impact on legumes yield is shown if figure B1.

As the carbon and nitrogen cycle are closely linked, the overall change in NBP is qualitatively be similar to that of the overall N balance (see Fig. S5). However, we agree that explicitly showing the main C balance components increases the informative value of the paper. We added a Figure in the appendix (Fig. C9) similar to Fig. 4 that shows the temporal evolution of the C balance components. It shows C input from manure, establishment and NPP as well as C losses from heterotrophic respiration, harvest and fire. We added two short paragraphs describing the results for the potential natural vegetation (L274-277) and dynamic land use (L297-300) simulations. The main differences are reduced NPP in the C-costly approach which is a result of the cost of BNF. This leads to lower biomass availability and in turn reduced losses through harvest, fire and heterotrophic respiration. We extended the discussion respectively (L358ff).

Also on figure 4 we see the relatively large impact of the new BNF parameterization on N emissions. It would then be interesting to show a comparison of these simulated fluxes to observations, as it is done for BNF. Especially for N2O emissions. It is obviously an important component of the GHG budget. So, with the new BNF parameterization, does it improve the simulated fluxes of N2O ?

We moved the table B1, which included global literature estimates, from the appendix into the main text (Table 2) and added the following additional literature estimates:$N_2O$ emission from Scheer et al. (2020) and volatilization from Bouwman et al. (2002) and Bouwman et al. (1997) and amended these to the results section (L268ff and 291ff) and the discussion (L369-374).

Minor remarks:

l 268: The authors seem surprised that the new approach does not limit the crop yield. But if I understood well the model, it is not so surprising for me. Since in condition when NPP is not a limiting factor for BNF (that should be the case for crops) and, as the model try to fulfill the N limitation, then the simulated BNF should be sufficient to fill the N demand of the plant and then should not produce N limitation ? Then it could explain why even if the different approaches give different BNF there is no impact of yield. This is exactly what we expect from the new formulation compared to original: define the BNF to avoid N limitation but without N excess... This is also the reason why it would be interesting to show the global impact on NPP: We should expect a decrease in NPP on carbon and N limited ecosystems, as the C-cost or N stress could be too high to be fulfilled by the BNF fixation. On the contrary, we should have no change in ecosystem with few limitations even is the BNF is reduced.

Thank you for bringing this up. While NPP may not be limiting for BNF, we expected it to be lower because of the investment cost for BNF, which is subtracted from NPP, thereby reducing the NPP available for plant growth and grain formation. We therefore expected a reduction in legume crop yield compared to the implementation in which the crops got all the N they need for free. We show that global yields of soybean and pulses are reduced (L248f in the preprint and L293f in the revised manuscript, and Fig S3 and S4). However, we expected a stronger reduction. We provided one explanation which is that the reduced respiratory losses of NPP balance expenses for BNF (L269-273 in the preprint and L340-344 in the revised manuscript, and Fig. B2).

Nevertheless, we agree that including the additional insights from the C balance assessment (see reply to major comment two) further improves the discussion of this aspect.

Figure 2: what are the percentage indicated in blue and red in a) ?

These indicate the overlap between simulated and observed ranges. We amended the caption of Fig. 2 and provide the equation for the overlap in the model evaluation section (L165-169).

Figure 3: the DBf term is not defined. I guess it is the observation, but it should be described

Thank you. This was also pointed out by reviewer 1 and is indeed referring to the Davies-Barndard and Friedlingstein data. We now included the explanation of DBF in the caption.

**Additional changes**

In addition to the changes we made following the reviewer comments, we also made the following additional changes to improve the quality of the manuscript:

In the submitted version of the manuscript we did not mention that the BNF also depends on the root distribution allowing shallow rooting plants to fix more N compared to deep rooting plants. We added a sentence on limitation of root distribution in the C-costly approach (L124ff).

We updated the panels e) and f) of Fig. C3 and C4 as these did show the agricultural BNF (including grasslands) per grid cell area and not per cropland area in the original version. Panels e) and f) now show the legume crop BNF (excluding grasslands) per legume cropland area. We are certain that reporting the values this way is more intuitive and that excluding grasslands is reasonable as we mainly discuss cropland BNF.

We updated Fig. C7 and C8 correcting a scaling error in the calculation of $\%N_{dfa}$ and indcluding a panel for the Original approach in Fig. C7. Correctly scaled, $\%N_{dfa}$ values compare substantially better to empirical data and adding the data for the Original approach highlights the advantages of the C-costly approach.

We corrected language errors in L95, L288 and L428

We updated equation formatting to comply with Copernicus standards and harmonized the use of abbreviations.

Changed the color scheme of Fig. B1 to match Fig. 2.

**Bibliography**

Bouwman, A.F., Boumans, L.J.M., Batjes, N.H., 2002. Estimation of global NH3 volatilization loss from synthetic fertilizers and animal manure applied to arable lands and grasslands. Global Biogeochemical Cycles 16, 8-1-8–14. https://doi.org/10.1029/2000GB001389

Bouwman, A.F., Lee, D.S., Asman, W. a. H., Dentener, F.J., Van Der Hoek, K.W., Olivier, J.G.J., 1997. A global high-resolution emission inventory for ammonia. Global Biogeochemical Cycles 11, 561–587. https://doi.org/10.1029/97GB02266

Frerebeau, N., Lebrun, B., Arel-Bundock, V., Stervbo, U., 2024. khroma: Colour Schemes for Scientific Data Visualization.

Halliday, J., Pate, J.S., 1976. The acetylene reduction assay as a means of studying nitrogen fixation in white clover under sward and laboratory conditions. Grass and Forage Science 31, 29–35. https://doi.org/10.1111/j.1365-2494.1976.tb01112.x

Herridge, D.F., Peoples, M.B., Boddey, R.M., 2008. Global inputs of biological nitrogen fixation in agricultural systems. Plant Soil 311, 1–18. https://doi.org/10.1007/s11104-008-9668-3

Jiang, S., Jardinaud, M.-F., Gao, J., Pecrix, Y., Wen, J., Mysore, K., Xu, P., Sanchez-Canizares, C., Ruan, Y., Li, Q., Zhu, M., Li, F., Wang, E., Poole, P.S., Gamas, P., Murray, J.D., 2021. NIN-like protein transcription factors regulate leghemoglobin genes in legume nodules. Science 374, 625–628. https://doi.org/10.1126/science.abg5945

Kaschuk, G., Kuyper, T.W., Leffelaar, P.A., Hungria, M., Giller, K.E., 2009. Are the rates of photosynthesis stimulated by the carbon sink strength of rhizobial and arbuscular mycorrhizal symbioses? Soil Biology and Biochemistry 41, 1233–1244. https://doi.org/10.1016/j.soilbio.2009.03.005

Kou-Giesbrecht, S., Arora, V.K., Seiler, C., Arneth, A., Falk, S., Jain, A.K., Joos, F., Kennedy, D., Knauer, J., Sitch, S., O'Sullivan, M., Pan, N., Sun, Q., Tian, H., Vuichard, N., Zaehle, S., 2023. Evaluating nitrogen cycling in terrestrial biosphere models: a disconnect between the carbon and nitrogen cycles. Earth Syst. Dynam. 14, 767–795. https://doi.org/10.5194/esd-14-767-2023

Liu, Y., Wu, L., Baddeley, J.A., Watson, C.A., 2011. Models of biological nitrogen fixation of legumes. A review. Agronomy Sust. Developm. 31, 155–172. https://doi.org/10.1051/agro/2010008

Meyer, D.R., Anderson, A.J., 1959. Temperature and Symbiotic Nitrogen Fixation. Nature 183, 61–61. https://doi.org/10.1038/183061a0

Montañez, A., Danso, S.K.A., Hardarson, G., 1995. The effect of temperature on nodulation and nitrogen fixation by five Bradyrhizobium japonicum strains. Applied Soil Ecology 2, 165–174. https://doi.org/10.1016/0929-1393(95)00052-M

Patterson, T.G., Larue, T.A., 1983. Root Respiration Associated with Nitrogenase Activity (C2H2) of Soybean, and a Comparison of Estimates 1. Plant Physiology 72, 701–705. https://doi.org/10.1104/pp.72.3.701

Reed, S.C., Cleveland, C.C., Townsend, A.R., 2011. Functional Ecology of Free-Living Nitrogen Fixation: A Contemporary Perspective. Annual Review of Ecology, Evolution, and Systematics 42, 489–512. https://doi.org/10.1146/annurev-ecolsys-102710-145034

Rousk, K., Sorensen, P.L., Michelsen, A., 2018. What drives biological nitrogen fixation in high arctic tundra: Moisture or temperature? Ecosphere 9, e02117. https://doi.org/10.1002/ecs2.2117

Ryle, G.J.A., Powell, C.E., Gordon, A.J., 1979. The Respiratory Costs of Nitrogen Fixation in Soyabean, Cowpea, and White Clover: I. Nitrogen fixation and the respiration of the nodulated root. Journal of Experimental Botany 30, 135–144. https://doi.org/10.1093/jxb/30.1.135

Salvagiotti, F., Cassman, K.G., Specht, J.E., Walters, D.T., Weiss, A., Dobermann, A., 2008. Nitrogen uptake, fixation and response to fertilizer N in soybeans: A review. Field Crops Research 108, 1–13. https://doi.org/10.1016/j.fcr.2008.03.001

Scheer, C., Fuchs, K., Pelster, D.E., Butterbach-Bahl, K., 2020. Estimating global terrestrial denitrification from measured N2O:(N2O + N2) product ratios. Current Opinion in Environmental Sustainability, Climate Change, Reactive Nitrogen, Food Security and Sustainable Agriculture 47, 72–80. https://doi.org/10.1016/j.cosust.2020.07.005

Serraj, R., Sinclair, T.R., Purcell, L.C., 1999. Symbiotic N2 fixation response to drought. Journal of Experimental Botany 50, 143–155. https://doi.org/10.1093/jxb/50.331.143

Valentine, A.J., Benedito, V.A., Kang, Y., 2018. Legume Nitrogen Fixation and Soil Abiotic Stress: From Physiology to Genomics and Beyond, in: Annual Plant Reviews Online. John Wiley & Sons, Ltd, pp. 207–248. https://doi.org/10.1002/9781119312994.apr0456

Voisin, A.S., Salon, C., Jeudy, C., Warembourg, F.R., 2003. Symbiotic N2 fixation activity in relation to C economy of Pisum sativum L. as a function of plant phenology. Journal of Experimental Botany 54, 2733–2744. https://doi.org/10.1093/jxb/erg290

von Bloh, W., Schaphoff, S., Müller, C., Rolinski, S., Waha, K., Zaehle, S., 2018. Implementing the Nitrogen cycle into the dynamic global vegetation, hydrology and crop growth model LPJmL (version 5). Geoscientific Model Development.

Wu, McGechan, 1999. Simulation of nitrogen uptake, fixation and leaching in a grass/white clover mixture. Grass and Forage Science 54, 30–41. https://doi.org/10.1046/j.1365-2494.1999.00145.x

Yao, Y., Han, B., Dong, X., Zhong, Y., Niu, S., Chen, X., Li, Z., 2024. Disentangling the variability of symbiotic nitrogen fixation rate and the controlling factors. Global Change Biology 30, e17206. https://doi.org/10.1111/gcb.17206